# Evolution of the Coupling Coordination between the Marine Economy and Digital Economy

Yang Liu [1] , Yiying Jiang [2,*], Zhaobin Pei [1,*], Na Xia [2] and Aijun Wang [1]

1 School of Marine Law and Humanities, Dalian Ocean University, Dalian 116023, China
2 School of Economics and Management, Dalian Ocean University, Dalian 116023, China
* Correspondence: jiangyiying@dlu.edu.cn (Y.J.); pzb@dlou.edu.cn (Z.P.)

**Abstract:** Accelerating the high-quality integrated development of digital economy and marine economy is vital for the development of the marine economy in coastal countries and regions. However, few scholars examined such coordination. Here, based on panel data from 2012 to 2019 and the spatial scale of China's coastal provinces and cities, the entropy method, coupling harmonious degree model (CCDM), Theil index, and Tobit model were adopted to measure and calculate the interval index differences in the marine economic quality and digital economy level. Exploring the coordination between the marine economy and digital economy, the interval difference index, and the coordination impact factors were also important. First, we found that the quality level of the marine economy and digital economy moved forward in waves and spiraled up, but that the quality of development was relatively low. Second, the coordination between the marine economy and digital economy gradually increased. Third, the coordination gap between the regional marine economy and digital economy was obvious. Fourth, the main factors that affected the coordination between the marine economy and digital economy were the level of digital infrastructure construction, the scale of the marine economy, the level of the marine industry, and industrial digitalization. The results have value for the sustainable development of the marine economy of coastal countries and regions.

**Keywords:** digital economy; marine economy; coordination; high quality

## 1. Introduction

As an important support of the national economy, the marine economy is an important carrier of opening up, an important guarantee of national economic security, and a strategic space for future development. The international community now generally believes that the marine economy is an important part of the global economy. As a new economic frontier and growth engine, the marine economy has great potential to stimulate economic growth, create employment and promote innovation [1]. Most of the world's megacities are located in the coastal zone. These port cities have a highly concentrated global population and assets, which are an important part of the global economy. And the importance of the port cities in international trade has increased significantly. According to the prediction of the Organization for Economic Cooperation and Development, by 2030, the marine industry with great development potential will outperform the overall performance of the global economy in terms of added value and job creation, and the contribution of the marine economy to the global economic added value will double to $3 trillion, accounting for about 2.5% of the global economic added value. (OECD, 2016) [2].

According to the Statistical Bulletin of China's Marine Economy in 2021, the national marine GDP exceeded CNY 9 trillion, reached CNY 9038.5 billion, and increased by 8.3% compared with the previous year. It contributed 8% to national economic growth and accounted for 15% of coastal GDP, which was 0.1% higher than the previous year's GDP. In the new stage of development, China's marine economy is developing well but is also facing many problems, such as a sloppy development model, insufficient growth momentum, a low level of industrial digitalization, weak scientific and technological innovation, a

poor coordination mechanism, and low governance capacity. The main reason for this is that the data elements or production factors of the marine economy have not been fully activated, the level of digital empowerment is low, and the digital economy has not yet been deeply integrated to form a linkage development situation. Thus, the digital economy is a new economic form and has become a new engine for sustainable economic growth in China [3]. Through the deep penetration of digital technology and the real economy, we can promote the birth of new industries, new formats, and new models; and strengthen the new engine of economic development [4]. This provides a new theoretical framework and realistic path for the high-quality development of the marine economy. In the new development environment, we can promote the coordination and integration of the marine economy and the digital economy, forming an endogenous growth mechanism of "mutuality and symbiosis, cooperation, and a win-win". It is conducive to giving full play to the synergistic effect of the marine economy and digital economy, transforming the development model of the marine economy, solving the problems of marine economic development, and comprehensively improving the quality of the marine economy and digital economy development. Therefore, it is of great theoretical significance and practical value to comprehensively study the coordination between the marine economy and digital economy. Previous research on the marine economy focused on the contribution of the marine economy to the national economy [5–7], the contributions of marine elements [8,9], and the economic differences between inland and coastal regions [10,11]. The digital economy is based on digital technologies. The qualitative research literature argues that the digital economy is multi-disciplinary [12]. In addition, the quantitative research literature is concerned with the appropriateness of traditional research approaches and methodologies that attempt to measure the development of the digital economy from different aspects [13–18]. The relationship between the marine economy and digital economy has become the current focus of scholars. Research on it mainly focuses on the development of marine digitalization and the influence of digital technology and data elements on the high-quality development of the marine economy. The relevant literature mainly covers various areas. Research on the development of ocean digitization is mainly based on the construction of a "digital ocean" [19–21]. With the rapid development of the digital economy, digital technology has had a profound impact on the development of the marine economy. Studies on the impact of digital technology and the digital economy on the marine economy have mostly been based on provincial and municipal panel data. They revealed the extent of the impact of digital technology and the digital economy on the development of the marine economy by building a comprehensive evaluation system [22–25] or theoretical analysis [26] that focused on the high-quality development of the marine industry, the efficiency of the marine economy, port logistics, marine fisheries, and marine cultural tourism. Currently, the importance of data elements is growing and the research perspective is turning to the role of data elements in the marine economy. Sun et al. constructed the input–output index system of data production factors and explored the value added by data factors in the process of the total factor productivity change of the marine economy [27]. For the coordinated development of the marine economy, domestic and foreign scholars conducted research based on the coordination of the marine economy with science and technology, finance, industry, and other areas [28–32]. Wang et al. used an impulse response to analyze the interactive relationship between the marine industry and marine science and technology [33]. Similarly, Ma et al. used the coordination degree model to comprehensively describe the coordinated evolution [34]. With the in-depth practice of green development, regional integration, and other concepts, we found that the research perspective gradually shifted to the marine economy and regional, ecological, high-quality, coordinated sea and land development [35–40]. Researchers mostly used theory, the coupling harmonious model, and the panel model for analysis and calculation. Based on analyzing the connotation of sustainable development of the marine economy, Martinez et al. evaluated its sustainable development ability from many aspects, such as marine science and technology, and discussed the practice of marine governance and

development [41]. Colazingari conducted research on the marine economy, marine science and technology, and marine resource conservation, and they elaborated on the relationship between marine resource development, marine economy development, and marine science and technology [42]. However, few scholars explored the coordination between the marine economy and digital economy.

In summary, most domestic and international scholars used relevant measurement tools to analyze the role of digital technology and data elements in promoting the development of the marine economy and the coordinated development of the marine economy with the environment, science and technology, finance, and the region. These research results had important reference value for this study. With the deep integration of the marine economy and digital economy, it is necessary to conduct in-depth research on the coordination of the marine economy and digital economy. Specifically, research has not been carried out on the state of the coordinated evolution between the marine economy and the digital economy, the spatial pattern of the evolution, the heterogeneity of the coordination between regions, and the factors that affect the coordination between the two. The comprehensive analysis of these problems is an urgent task for the high-quality development of the marine economy and digital economy in coastal countries and regions in the new era. As such, we took coastal provinces (cities) as the spatial scale, deeply analyzed the coordination relationship between the marine economy and digital economy, revealed the dynamic evolution law of their coordination, and constructed a panel econometric model to explore the key driving factors of the coordination between the marine economy and digital economy. The aim was to provide a reference for coastal countries and regions in the world to formulate policies to improve the coordination between the marine economy and digital economy. This paper makes contributions in the following aspects. First, from the theoretical level, we discuss the relationship between marine economic development and the digital economy and enrich the literature on the sustainable development of the marine economy. Second, from the empirical perspective, it presents an analysis of the characteristics of the coordinated space-time evolution and regional differences in the marine economy and digital economy, and it reveals the dynamic adjustment process of their coordination. Third, choosing the major coastal cities in China as the example, this paper discusses the coordinated suggestions for improving the marine economy and digital economy from the aspects of industrial coordination, infrastructure connectivity, and ecological coordinated governance.

## 2. Theoretical Analysis and Research Methods

### 2.1. Theoretical Analysis

The ocean is a valuable asset for sustainable development and is of great significance to human survival and development. And the marine economy has become one of the most dynamic and promising areas for economic growth in coastal countries. According to relevant statistics, the scale of the global digital economy has reached 32.2 trillion US dollars, accounting for 47.8% of GDP. The digital economy has become a new driving force for the economic growth of major global economies. As pointed out in the "2020 Global Competition Report" released by the World Economic Forum in 2021, data has become one of the most important and critical production factors in today's society. Digital economy is a higher economic stage after agricultural economy and industrial economy. The digital economy is a new economic form that takes digital knowledge and information as the key production factor, digital technology innovation as the core driving force, modern information network as the important carrier, through the deep integration of digital technology and the real economy, continuously improves the level of digitalization and intelligence of traditional industries, and accelerates the reconstruction of economic development and government governance model [43]. As a new economic form, it was fully launched in the world and has become the core driving force for economic growth.

According to the National Standard of the People's Republic of China "National Economic Industry Classification" (GB/T4754-2002) and the Marine Industry Standard

of the People's Republic of China "Marine Economic Statistical Classification and Code" (HY/T052-1999), the marine primary industry includes marine fisheries; the marine secondary industry includes marine oil and gas industry, coastal sand mining, the marine salt industry, the marine chemical industry, the marine biomedical industry, the marine power and seawater utilization industry, the marine ship industry, and the marine engineering construction industry; and the marine tertiary industry includes the marine transportation industry, coastal tourism, marine scientific research, education, and social services. In a narrow sense, the marine economy includes five areas: production and services that obtain products directly from the sea; primary processing of the former; production and services that are directly applied to marine development; production and services based on seawater or the sea; and a range of marine-related scientific research, education, and management. In a broad sense, marine industrial activities also include upstream and downstream industrial activities related to the marine economy in a narrow sense, regardless of whether the industry is located on the coast or not, as long as it provides certain conditions and a basis for the development and use of the sea.

Meanwhile, the digital economy comprises the communication industry, the computer basic technology industry, the software industry, the software integration industry, and the Internet industry, among five other industries. The digital economy includes digital industrialization and industry digitization. First, digital industrialization includes the electronic information manufacturing industry, the telecommunications industry, the software and digital service industry, and the Internet industry. Second, industrial digitalization means that traditional industries benefit from the improvement of production quantity and quality brought about by digital technology. On this basis, the integration of various fields will be realized, resulting in innovation modes and forms. The digital economy development should take new infrastructure as the core underlying architecture, including semiconductor equipment, communication facilities and services, software applications, and other basic software and hardware. On this foundation, various Internet enterprises contact the specific application scenarios of enterprises and individuals and build relevant digital application platforms to meet the specific needs of customers; in addition, the new output generated by the three traditional industries using digital technology is a direct manifestation of digital empowerment.

The marine economy and digital economy are two subsystems of the social economic system. From the perspective of system theory, the two have the characteristics of openness and organicity. They are not zero-sum games and contradict each other, but are mutually coupled and mutually beneficial. With the development and penetration of digital technology, the relationship between them is more close, complex, and diversified. A one-sided emphasis on the digital economy or the marine economy will lead to resource surplus and development lag. Only by realizing complementary advantages and dynamic optimization can the two promote the development from a low to a high level, and realize the situation of "mutual benefit, symbiosis and win-win cooperation". The coordination relationship between the marine economy and digital economy is shown in Figure 1.

First, the marine economy provides the material basis for the digital economy. It is widely known that the marine economy produced a large amount of data in the process of ecological governance, industrial development, scientific and technological innovation, and scale expansion. These data are important data elements for the development of the digital economy and the basis for digital economy development. The sustainable development of the marine economy can create more new application scenarios for the development of the digital economy; provide important application objects for the innovative development of digital technology; and provide broad space for the sustainable development of the digital economy, such as market, talent, resources, and information, and become the spatial carrier of the development of the digital economy. The sustained development of the marine economy will generate high-end and diversified demand, which will promote the development of technological innovation in the digital economy and promote the improvement of the innovation capacity of the digital economy.

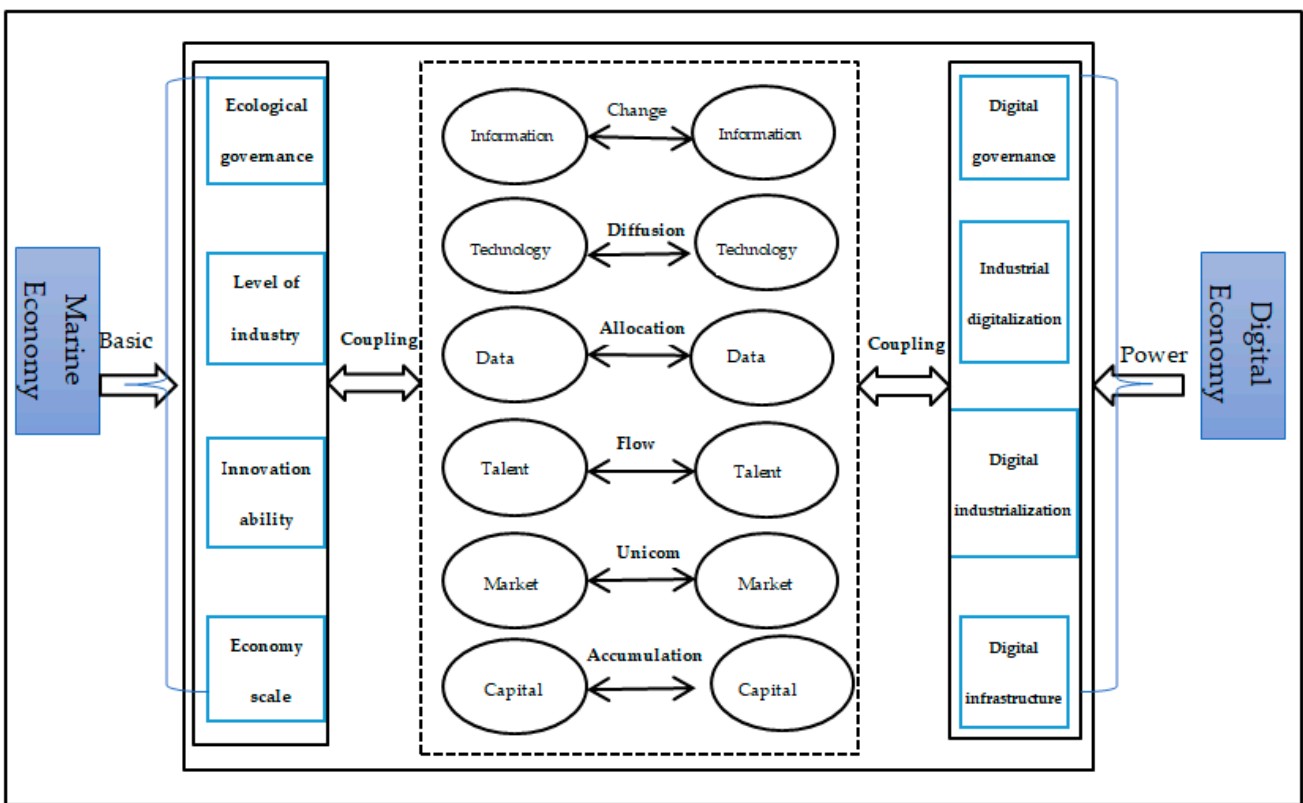

**Figure 1.** Coupling coordination relationship between the marine economy and digital economy.

Second, the digital economy provides a new impetus for the marine economy development. Also, the digital economy can empower the marine economy, promote the optimization and upgrading of the marine industrial structure, grow new vitality, form new growth points, and improve the resilience and development quality of the marine economy. The digital economy uses data elements to lead the free flow and reorganization of technology, capital, and material flows of the marine economy, and build a modern characteristic marine industry system. Meanwhile, the digital economy itself also spawned more new industries and new formats to promote the upgrading and expansion of the marine economy. In addition, the digital economy can accurately identify the real needs of the marine economy industry chain; effectively avoid excessive energy consumption, environmental pollution, and other problems in marine industry development; and also promote the development of marine industry towards high-end, green, cluster and intelligent. Digital industrialization can lead the optimization and reorganization of marine economic factors with data flow, break the space-time constraints of the flow of production factors, realize the free flow of marine economic production factors, promote innovative products and services, and reshape the marine economy pattern.

Third, the marine economy and the digital economy interact and co-exist. The marine economy and the digital economy realize deep integration through information, technology, data, talent, market, and capital, among other factors. The various production factors interact with each other; reorganize the factor resources in dynamic adjustment; optimize the production, circulation, consumption, and distribution of the marine economy and the digital economy; enhance the momentum of high-quality development; improve the quality and efficiency of development; realize the cooperation and symbiosis between the two; and constantly tend toward a more advanced state of coordination.

### 2.2. Indicator System Construction

This study took 11 coastal provinces (cities) in China (i.e., Liaoning, Tianjin, Hebei, Shandong, Jiangsu, Shanghai, Zhejiang, Fujian, Guangdong, Guangxi, and Hainan) as the research objects (excluding Hong Kong, Macao, and Taiwan). We measured the level of the marine economy and digital economy and focused on exploring the coordination relationship between the two. In China, scholars' research on marine economy is basically decentralized [44]. According to the definition of the national standard Classification of Marine and Related Industries (GB/T 20794-2006), the marine economy is the sum of all kinds of industrial activities for the development, utilization and protection of the marine and their associated activities. From the perspective of industry classification, marine economy includes marine industry and marine related industries. Marine industry refers to the production and service activities carried out in the development, utilization and protection of the sea. Marine related industries refer to the upstream and downstream industries that take various inputs and outputs as the link and form technical and economic links with major marine industries [45]. The digital economy, which is a new economic form with the development of the Internet, its concept can be traced back to the 1990s [46]. The G20 Digital Economy Development and Cooperation Initiative defines "digital economy" as: "A series of economic activities with the use of digital knowledge and information as key production factors, modern information network as an important carrier, and the effective use of information and communication technology as an important driving force for efficiency improvement and economic structure optimization" [47]. The development of the digital economy is integrating resources globally, shaping the economic structure globally, and transforming the competitive situation globally. We should give full play to the advantages of big data and multi-type application scenarios, promote the integration of digital technology into the real economy, accelerate the transformation and upgrading of the development of traditional industries, and advance into innovative industries, innovative formats, and innovative models [48]. The sound development of the digital economy will open up a new situation. In the process of digitalization, resources will be integrated, markets will be integrated, industrial model restructuring will be accelerated, cross-border development will be achieved, space restrictions will be broken, and industrial chain will be enriched. The digital economy promotes a sound economic system, accelerates the transformation of traditional industries, and forms the main driving force for the development of a modern economic system [49]. All data were selected from the China Ocean Yearbook, China Marine Statistical Yearbook, the China Statistical Yearbook, China Information Industry Yearbook, and China Industrial Economy Statistical Yearbook from 2012 to 2019 to ensure temporal and spatial consistency in measurements (the Ministry of Natural Resources divides the national marine economy into three circles: the northern marine economy circle(NMEC), mainly including Liaoning, Hebei, Tianjin, and Shandong; the eastern marine economy circle(EMEC), mainly including Jiangsu, Shanghai, and Zhejiang; and the southern marine economy circle(SMEC), mainly including Fujian, Guangdong, Guangxi, and Hainan).

To accurately measure the coordination between the marine economy and the digital economy, we divided the level of the marine economy into four dimensions: ecological environment, industrial level, economy scale, and innovation capacity. Five indicators were selected for each dimension, for a total of 20 measurement indicators to form the evaluation system of the level of the marine economy. As the development level of the digital economy is mainly classified using the four dimensions of digital governance, industry digitalization, digital industrialization, and digital infrastructure, and each dimension had five indicators, this gave a total of 20 measurement indicators that were used to build the evaluation index system of the digital economy level, as shown in Table 1.

**Table 1.** China's marine economy and digital economy level evaluation index system.

| Target Layer | Rule Layer | Weight | Index Layer | Index (Positive/Negative) | Weight |
|---|---|---|---|---|---|
| Level of marine economy | Ecological environment | 0.1246 | Volume of direct marine wastewater discharge (billion tons) | Negative | 0.0067 |
| | | | Relative perennial change in sea level (mm) | Negative | 0.0175 |
| | | | Proportion of near-shore category I and II water quality (%) | Positive | 0.0228 |
| | | | Wetland area (thousand hectares) | Positive | 0.0437 |
| | | | Nearshore and coastal area (km$^2$) | Positive | 0.0339 |
| | Industrial level | 0.2373 | Total output value of marine economy (billion CNY) | Positive | 0.0529 |
| | | | Tertiary industry output value (billion CNY) | Positive | 0.0555 |
| | | | Marine-related industries (billion CNY) | Positive | 0.0485 |
| | | | Fishermen's net income per capita (CNY) | Positive | 0.0225 |
| | | | Output of marine aquatic products (billion tons) | Positive | 0.0581 |
| | Economic scale | 0.3169 | Ocean freight volume (million tons) | Positive | 0.0639 |
| | | | Port cargo throughput (million tons) | Positive | 0.0437 |
| | | | Passenger throughput (10,000 people) | Positive | 0.0937 |
| | | | Standard container throughput (million TEU) | Positive | 0.0633 |
| | | | Amount of coastal tourism (million people) | Positive | 0.0522 |
| | Innovation capability | 0.3211 | Number of marine research employees (people) | Positive | 0.0428 |
| | | | Investment in marine scientific research (million CNY) | Positive | 0.0544 |
| | | | Number of scientific research projects (items) | Positive | 0.0674 |
| | | | Number of patents granted (pieces) | Positive | 0.0921 |
| | | | Published scientific and technical papers (pieces) | Positive | 0.0644 |
| Level of digital economy | Digital governance | 0.2371 | Information transmission, software, and information technology service industry fixed asset investment (billion CNY) | Positive | 0.0337 |
| | | | Number of industrial enterprises above the scale of R&D projects (items) | Positive | |
| | | | Number of digital economy enterprises (pieces) | Positive | 0.0538 |
| | | | Total turnover of technology contracts (million CNY) | Positive | |
| | | | Number of patent applications granted (pieces) | Positive | 0.0344 0.0572 0.0581 |
| | Industry digitization | 0.2824 | Online retail sales (billion CNY) | Positive | 0.0747 |
| | | | Number of business transaction activities enterprises (pieces) | Positive | 0.0461 |
| | | | E-commerce sales (billion CNY) | Positive | 0.0561 |
| | | | Digital Financial Inclusion Index | Negative | 0.0167 |
| | | | Express volume (million pieces) | Positive | 0.0891 |

**Table 1.** *Cont.*

| Target Layer | Rule Layer | Weight | Index Layer | Index (Positive/Negative) | Weight |
|---|---|---|---|---|---|
| Level of digital economy | Digital industrialization | 0.3156 | Number of employees in software and information services (people) | Positive | 0.0525 |
| | | | Income from information technology consulting services (million CNY) | Positive | 0.0673 |
| | | | Telecommunications business volume (billion CNY) | Positive | 0.0622 |
| | | | Electronic information manufacturing revenue (billion CNY) | Positive | 0.0803 |
| | | | Software industry revenue (billion CNY) | Positive | 0.0533 |
| | Digital infrastructure | 0.1648 | Number of Internet domain names (million names) | Positive | 0.0605 |
| | | | Number of Internet broadband access ports (million ports) | Positive | 0.0311 |
| | | | Telephone penetration rate (units per 100 people) | Positive | 0.0123 |
| | | | Length of long-distance fiber optic cable lines (km) | Positive | 0.0278 |
| | | | Internet broadband access users (million users) | Positive | 0.0311 |

Note: The weights in the table were obtained according to the research method presented in Section 2.3.1.

### 2.3. Research Methods

In this study, based on the definition of the digital economy and marine economy, theoretical foundation, and the current development situation, we selected the entropy method, coupling harmonious degree model (CCDM), Theil index, and Tobit model.

#### 2.3.1. Entropy Method

The concept of entropy has been predominantly applied in thermodynamics. In 1965, Zadeh was the first to introduce the concept of entropy into analytical decision-making, using entropy values to describe the ambiguity of information [50]. Burillo and Bustince (1996) argued that the degree of uncertainty of intuitionistic fuzzy sets can be measured using intuitionistic fuzzy entropy [51]. The entropy method is an objective weighting method used to determine the weight of indicators according to the size of the information provided by the observed values of each indicator. The advantage of entropy weight method is that it can ensure that the index weight is not affected by subjective factors when determining the index weight. The index weight can be calculated according to specific formulas and company data, making the evaluation results more objective, accurate and scientific. In this research, we selected the entropy method to calculate the comprehensive scores of each indicator of China's marine economy and digital economy levels. The calculation steps were as follows.

The index entropy calculation:

$$e_j = -k \sum_{i=1}^{n} p_{ij} \ln(p_{ij}) \tag{1}$$

In the model, $p_{ij} = A_{ij} / \sum_{i=1}^{n} A_{ij}$, $e_j$ represents the index entropy ($0 \leq e_j \leq 1$), $n$ represents the number of indexes, $k = 1/\ln m (k > 0)$, and $m$ represents the number of evaluation objects.

Index weight determination:

$$w_j = (1 - e_{ij}) / \sum_{i=1}^{n} (1 - e_{ij}) \tag{2}$$

In the model, $w_{ij}$ represents the index weight, $e_{ij}$ represents the index entropy, and $w'_j = \sum_{i=1}^{m} w_j$ represents the rule layer weight.

Calculation of the comprehensive score:

$$S = \sum_{j=1}^{n} w_j A_{ij} \tag{3}$$

2.3.2. CCDM

Coupling, also known as coupling degree, is a measure of the degree of correlation between modules. The strength of coupling depends on the complexity of the interface between modules, the way of calling modules and the amount of data transmitted through the interface. The coupling degree between modules refers to the dependency relationship between modules, including control relationship, call relationship and data transfer relationship [52]. Coupling is now widely used in studies of changes between elements. First, the method revealed the current average development level of China's marine economy and digital economy. Second, it identified the indicators that made the greatest contribution to the two systems in the CCDM. Third, it evaluated the current level and development of the coupling of the marine economy and digital economy. Fourth, it explored different influences on the parameters of the coupling model in different provinces. The specific formula is as follows:

$$C_{ij} = \sqrt{S_{ijr} \times S_{ijt}} / (S_{ijr} + S_{ijt}) \tag{4}$$

In the model, $C_{ij}$ represents the coupling value ($0 \leq C_{ij} \leq 1$). The larger the number, the higher the coupling. $S_{ijr}$ represents the development level of the marine economy and $S_{ijt}$ represents the development level of the digital economy. To further calculate the coordination between China's marine economy and digital economy in detail, we established the following model:

$$T_{ij} = \alpha S_{ijr} + \beta S_{ijt} \tag{5}$$

$$D_{ij} = \sqrt{C_{ij} \times T_{ij}} \tag{6}$$

In the model, $T_{ij}$ represents the comprehensive coordination index between the marine economy and digital economy of province $j$ in year $i$. $D_{ij}$ represents the coordination value between the marine economy and digital economy of province $j$ in year $i$. $D_{ij} \in [0,1]$ was obtained according to the Formula (6). $\alpha$ represents the marine economy weight, $\beta$ represents the weight of the digital economy, and the contributions of the two systems should be the same. Therefore, $\alpha = \beta = 0.5$. The value of $D_{ij}$ reflects the relationship between the two systems. The larger the value, the higher the degree of coordination between the marine economy and digital economy. The converse is also true. To reflect the coordination relationship between the two more intuitively, we divided the coordination grade of the two systems based on the existing classification methods, as shown in Table 2.

**Table 2.** Evaluation standard of coordination grade.

| Coordination Grade | RHC | Coordination Grade | RHC |
| --- | --- | --- | --- |
| 0.0 < D ≤ 0.2 | Severely disorders | 0.4 < D ≤ 0.6 | Primary coordination |
| 0.2 < D ≤ 0.3 | Mild disorders | 0.6 < D ≤ 0.8 | Intermediate coordination |
| 0.3 < D ≤ 0.4 | Barely coordination | 0.8 < D ≤ 1.0 | Senior coordination |

Note: RHC—rank of harmony coefficient.

2.3.3. Theil Index

Theil index or Theil entropy standard is named after Theil uses the concept of entropy in information theory to calculate income inequality. It was first proposed by Theil and Henri in 1967. Theil index was initially used to evaluate income equity, and has been

widely used in the research of equity between regions in recent years [53,54]. The scale of the Thiel index indicates the size of the distribution difference of the factors studied in different regions. The smaller the Thiel index, the smaller the difference, and vice versa. By analyzing the time series of the Theil index, we can clearly see the dynamic change process of the differences each year. The Theil index has been widely used in many fields, such as ecology, management, and society, to analyze regional differences. China's marine economy is mainly distributed in three major marine economy circles, with regional differences in the development levels. Therefore, we used the Theil index to measure the difference between the coordination of the marine economy and digital economy in China's coastal provinces. The specific formulas are as follows:

$$T = T_b + T_w \tag{7}$$

$$T_b = \sum_{k=1}^{K} y_k \log \frac{y_k}{n_k/n} \tag{8}$$

$$T_w = \sum_{k=1}^{K} y_k \left( \sum_{i \in g_k} \frac{y_i}{y_k} \log \frac{y_i/y_k}{1/n_k} \right) \tag{9}$$

In the model, $T_b$ and $T_w$ represent the inter-group gap and intra-group gap, respectively. The coordination of $n$ regional marine economy and digital economy is divided into $K$ clusters, each group is $g_k (k = 1, 2, \cdots, K)$, and the number of provinces in the $k$ group $g_k$ is $n_k$; $y_i$ and $y_k$ represent the coordination of provinces $i$ and the coordination of group $K$, respectively.

### 2.3.4. Tobit Model

Tobit model is an econometric model first proposed by James Tobin, an economist and winner of the Nobel Prize in Economics, when studying the demand for durable consumer goods in 1958 [55]. The characteristic of Tobit model is that it contains two parts: one is the choice equation model which represents the constraint conditions; the other is a continuous variable equation model under constraint conditions. The coordination between the marine economy and digital economy is characterized by a random distribution, and the value is between 0 and 1. If the ordinary least-squares (OLS) method were used for regression, it would have been unable to obtain a consistent estimate and the conclusion would be biased. Therefore, we used the maximum likelihood intercept regression model, that is, the Tobit model. The formula of the Tobit model is as follows:

$$Y = \begin{cases} a + \beta X_{it} + u_i + e_{it}, & Y > 0, \forall i, t \\ 0, & Y < 0, \forall i, t \end{cases} \tag{10}$$

In the model, Y is the coordination value vector of the marine economy and the digital economy, $X$ is the independent variable vector, a is the intercept term, $\beta$ is the parameter vector, $u$ is the random variable, and $e$ is the residual.

## 3. Spatial-Temporal Evolution of the Marine Economy and Digital Economy

Depending on the entropy method formula to calculate the marine economy quality index in China's coastal provinces (Equations (1)–(3)), we obtained the development levels of the marine economies in China's coastal provinces from 2012 to 2019. Table 3 shows the detailed results.

### 3.1. Spatial-Temporal Evolution of the Marine Economy Level

Considering time evolution, China's marine economy level showed a wave-like upward trend, which indicates that with the in-depth implementation of the marine power strategy, the marine economy gradually entered a benign development track. Moreover, the development level steadily improved and economic resilience gradually increased.

Among them, from 2012 to 2015, the marine economy development showed a straight upward trend, with an increase of more than 20%, especially in Liaoning, Shandong, Jiangsu, Shanghai, and Guangdong. From 2012 to 2015, the marine economy experienced major fluctuations. The marine economy levels of Liaoning, Tianjin, Hebei, Jiangsu, Shanghai, and other provinces and cities declined obviously, and the marine economy level of Liaoning declined the most, reaching 15%. From 2017 to 2019, except for Tianjin and Liaoning, the marine economy of the remaining provinces and cities showed a straight upward trend, with a growth rate of more than 10%. This indicates that the marine economy entered a stable growth period. However, in terms of the average value of marine economic development, China's marine economic level was still low, with the overall average value being only 0.3. Thus, it is urgent to choose a diversified development path to comprehensively improve the quality of marine economic development.

**Table 3.** Marine economy quality index in China's coastal provinces/cities (2012–2019).

| Province/City | 2012 | 2013 | 2014 | 2015 | 2016 | 2017 | 2018 | 2019 | Mean Value |
|---|---|---|---|---|---|---|---|---|---|
| Liaoning | 0.2398 | 0.2570 | 0.2798 | 0.3323 | 0.2792 | 0.2877 | 0.2837 | 0.2991 | 0.2823 |
| Tianjin | 0.1408 | 0.1548 | 0.1744 | 0.1822 | 0.1686 | 0.1790 | 0.1681 | 0.1863 | 0.1693 |
| Hebei | 0.0965 | 0.0999 | 0.1056 | 0.1114 | 0.1130 | 0.1289 | 0.1438 | 0.1845 | 0.1230 |
| Shandong | 0.4187 | 0.4445 | 0.4809 | 0.5028 | 0.5069 | 0.5315 | 0.6021 | 0.6592 | 0.5183 |
| Jiangsu | 0.2360 | 0.2757 | 0.2953 | 0.3104 | 0.2978 | 0.3134 | 0.3213 | 0.3645 | 0.3018 |
| Shanghai | 0.2945 | 0.3263 | 0.3418 | 0.3652 | 0.3266 | 0.3416 | 0.3572 | 0.3867 | 0.3425 |
| Zhejiang | 0.2673 | 0.2961 | 0.3115 | 0.3338 | 0.3438 | 0.3788 | 0.4122 | 0.4420 | 0.3482 |
| Fujian | 0.2096 | 0.2362 | 0.2559 | 0.2728 | 0.2865 | 0.3136 | 0.3413 | 0.3635 | 0.2849 |
| Guangdong | 0.4672 | 0.4924 | 0.5403 | 0.6462 | 0.6550 | 0.7362 | 0.8036 | 0.8881 | 0.6536 |
| Guangxi | 0.0079 | 0.0866 | 0.0943 | 0.1009 | 0.1026 | 0.1029 | 0.1324 | 0.1345 | 0.1042 |
| Hainan | 0.0885 | 0.0908 | 0.1007 | 0.1049 | 0.1134 | 0.1125 | 0.1358 | 0.1489 | 0.1119 |
| Mean value | 0.2309 | 0.2509 | 0.2710 | 0.2966 | 0.2903 | 0.3115 | 0.3365 | 0.3688 | 0.2945 |

Source: Calculated by the author according to the data.

Considering spatial evolution, the overall marine economy level presents a distribution pattern of "high in the East and low in the South and North". Among them, the average marine economy level in the east reached 0.33, and the level of the marine economy in each province was relatively balanced. The average marine economy level in the Northern Marine Economy Circle reached 0.27. Among them, Shandong had the highest level of marine economy, with an average of 0.52, which was four times that of Hebei. This indicates that Shandong's marine economy had a high priority and did not form a cluster development state with other provinces. The average marine economy level in the Southern Marine Economy Circle reached 0.28. Among them, Guangdong had the highest level of marine economy, with an average of 0.65, which was six times the average of Guangxi and Hainan. Also, there was a large gap in the level of the marine economy in the whole region and in the level of the marine economy between provinces. Guangdong had the highest level of marine economy, reaching 0.89 in 2019, followed by Shandong, with 0.66 in 2019. However, the marine economy levels of Hainan and Guangxi were relatively low, with averages of only approximately 0.1. In 2019, the marine economy levels of the two provinces were only 0.14, indicating that the level of China's marine economy was polarized in space and that the development was extremely uneven.

### 3.2. Spatial-Temporal Evolution of Digital Economy Level

Through the comprehensive calculation of the development levels of digital economies in China's coastal provinces and cities, we obtained the results shown in Table 4.

Considering time evolution, from 2012 to 2019, the digital economy of China's coastal provinces and cities as a whole showed a straight upward trend and grew rapidly. The average development of the digital economy rose from 0.09 to 0.22, with an average annual growth of 7.5%, reflecting the good development momentum of the digital economy in

coastal provinces and cities of China. Combined with each specific development stage, the development level of the digital economy in coastal provinces and cities increased in a stepwise manner from 2012 to 2016, especially in southern provinces and cities of China, such as Zhejiang, Guangdong, and Jiangsu. This was mainly because of the early development and relative strength of the digital economy in the southern cities of China. From 2016 to 2019, the digital economy development of coastal provinces and cities showed a vertical upward development trend, and the growth rate was significantly accelerated, higher than the growth rate from 2012 to 2016. This was because of the accelerated integration of China's digital technology and various industries, along with the gradual shift of the digital economy to a new stage of deepening application, standardized development, and inclusive sharing. As for the development quality, the overall digital economy development quality in coastal provinces and cities was relatively low. In 2019, the average value of digital economy development in coastal provinces and cities was only 0.2181, and only Guangdong reached more than 0.5, which indicates that China's coastal provinces and cities had low digital economic strength and still have the potential for growth. Thus, we should accelerate the deep integration with other industries and improve digital empowerment level.

**Table 4.** Level of digital economy development in coastal provinces (cities) (2012–2019).

| Province/City | 2012 | 2013 | 2014 | 2015 | 2016 | 2017 | 2018 | 2019 | Mean Value |
|---|---|---|---|---|---|---|---|---|---|
| Liaoning | 0.0933 | 0.1046 | 0.1270 | 0.1316 | 0.1181 | 0.1237 | 0.1278 | 0.1494 | 0.1225 |
| Tianjin | 0.0458 | 0.0519 | 0.0674 | 0.0801 | 0.0908 | 0.0931 | 0.1071 | 0.1283 | 0.0825 |
| Hebei | 0.0567 | 0.0664 | 0.0759 | 0.0912 | 0.1117 | 0.1261 | 0.1467 | 0.1822 | 0.1075 |
| Shandong | 0.1422 | 0.1883 | 0.2094 | 0.2413 | 0.2769 | 0.3002 | 0.3578 | 0.3815 | 0.2625 |
| Jiangsu | 0.2563 | 0.2993 | 0.3585 | 0.4174 | 0.4640 | 0.4808 | 0.5368 | 0.5969 | 0.4263 |
| Shanghai | 0.1267 | 0.1396 | 0.1789 | 0.2077 | 0.2343 | 0.2504 | 0.2884 | 0.3282 | 0.2188 |
| Zhejiang | 0.1832 | 0.2068 | 0.2370 | 0.3031 | 0.3551 | 0.3935 | 0.4613 | 0.5424 | 0.3350 |
| Fujian | 0.0872 | 0.0967 | 0.1154 | 0.1501 | 0.1928 | 0.2362 | 0.2618 | 0.2802 | 0.1775 |
| Guangdong | 0.3080 | 0.3586 | 0.4229 | 0.4999 | 0.5760 | 0.6647 | 0.7954 | 0.9558 | 0.5725 |
| Guangxi | 0.0347 | 0.0426 | 0.0512 | 0.0612 | 0.0728 | 0.0800 | 0.0977 | 0.1241 | 0.0700 |
| Hainan | 0.0061 | 0.0112 | 0.0141 | 0.0213 | 0.0231 | 0.0299 | 0.0348 | 0.0430 | 0.0225 |
| Mean value | 0.0909 | 0.1427 | 0.1691 | 0.2000 | 0.2291 | 0.2527 | 0.2927 | 0.3373 | 0.2181 |

Source: Calculated by the author according to the data.

Considering spatial evolution, the development level of the digital economy in China's coastal provinces and cities showed a distribution state of "high in the East and low in the South and North". The average level of the digital economy of the Eastern Marine Economy Circle reached 0.32. Among them, the highest average level of the digital economy in Jiangsu reached 0.43, while the average level of the digital economy in Shanghai reached 0.22. There was a small gap in the levels of digital economies between provinces. The average level of the digital economy of the Northern Marine Economy Circle was 0.14. Among them, Shandong had the highest level of digital economy, with an average of 0.26, which was four times that of Hebei. This shows that Shandong's marine economy had a high priority and did not form a cluster development state with other provinces. The average value of the digital economy in the Southern Marine Economy Circle reached 0.21. Among them, Guangdong had the highest level of digital economy, with an average of 0.57. Moreover, the development level of the digital economy between provinces was different and was increasing year by year. Guangdong, Zhejiang, and Jiangsu were in the leading positions and were significantly higher than other provinces and cities, while Guangxi and Hainan had the lowest average levels of development. Among them, the average development of the digital economy in Guangdong was about 28 times that of Hainan, indicating that the development of the digital economy between provinces was extremely uneven and that the polarization was relatively serious.

## 4. Coordinated Development of Marine Economy and Digital Economy

Relied on the CCDM formula, we have fully and accurately calculated the coordination degrees between the marine economy and digital economy in the coastal provinces from 2012 to 2019. The results are shown in Table 5 and Figure 2.

**Table 5.** Coordination between the marine economy and the digital economy in coastal provinces (2012–2019).

| Province/ City | 2012 | 2013 | 2014 | 2015 | 2016 | 2017 | 2018 | 2019 | Mean Value |
|---|---|---|---|---|---|---|---|---|---|
| Liaoning | 0.2735 | 0.2863 | 0.3070 | 0.3234 | 0.3013 | 0.3071 | 0.3086 | 0.3251 | 0.3040 |
| Tianjin | 0.2004 | 0.2117 | 0.2328 | 0.2458 | 0.2487 | 0.2540 | 0.2590 | 0.2781 | 0.2413 |
| Hebei | 0.1923 | 0.2018 | 0.2116 | 0.2245 | 0.2371 | 0.2525 | 0.2695 | 0.3028 | 0.2365 |
| Shandong | 0.3493 | 0.3803 | 0.3984 | 0.4173 | 0.4328 | 0.4469 | 0.4817 | 0.5007 | 0.4259 |
| Jiangsu | 0.3507 | 0.3790 | 0.4034 | 0.4242 | 0.4311 | 0.4406 | 0.4557 | 0.4830 | 0.4210 |
| Shanghai | 0.3108 | 0.3267 | 0.3516 | 0.3711 | 0.3719 | 0.3824 | 0.4006 | 0.4220 | 0.3671 |
| Zhejiang | 0.3326 | 0.3518 | 0.3686 | 0.3988 | 0.4180 | 0.4394 | 0.4669 | 0.4948 | 0.4089 |
| Fujian | 0.2600 | 0.2749 | 0.2932 | 0.3181 | 0.3428 | 0.3689 | 0.3866 | 0.3994 | 0.3305 |
| Guangdong | 0.4355 | 0.4584 | 0.4889 | 0.5331 | 0.5542 | 0.5914 | 0.6323 | 0.6787 | 0.5466 |
| Guangxi | 0.1618 | 0.1742 | 0.1864 | 0.1982 | 0.2079 | 0.2130 | 0.2385 | 0.2542 | 0.2043 |
| Hainan | 0.1079 | 0.1263 | 0.1373 | 0.1537 | 0.1600 | 0.1703 | 0.1854 | 0.2000 | 0.1551 |
| Mean value | 0.2704 | 0.2883 | 0.3072 | 0.3280 | 0.3369 | 0.3515 | 0.3713 | 0.3944 | 0.3311 |

Note: The results were calculated using the author's formula.

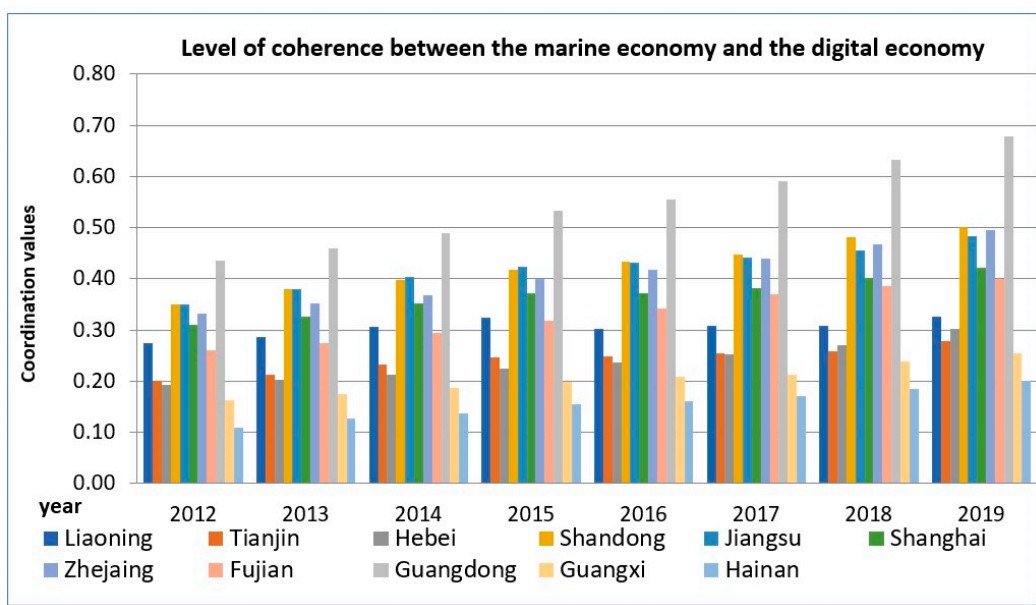

**Figure 2.** Coordination between the marine economy and the digital economy in China's coastal provinces.

### 4.1. Temporal Evolution of the Coordination between the Marine Economy and Digital Economy

Table 5 and Figure 2 show that the coordination between the marine economy and digital economy of the coastal provinces and cities increased steadily from 2012 to 2019. This indicates that the improvement of the marine economy over the 8 years studied was inseparable from the digital economy development. The digital economy has become the engine and new driving force of marine economy development. Still, at the level of coordinated development, the overall level was low. In 2019, the mean coordination of the whole was only 0.331, with Guangdong having the highest coordination average of 0.5466. The coordination average of Tianjin, Hebei, Guangxi, Hainan, and other provinces and cities was far lower than the overall average, while the coordination average of other provinces was mostly around 0.4.

According to Tables 3–5, the development and fluctuation trends of the marine economy and digital economy of coastal provinces and cities in China were not synchronized. During the study period, the marine economy developed slowly and fluctuated greatly, and some provinces and cities even experienced a downward trend. This was mainly because China's marine economy was in the period of optimizing and adjusting the industrial structure and accelerating the conversion of new and old kinetic energy, and the marine industry modernization system was being established. However, the development of the digital economy had a small fluctuation, and the overall development trend occurred in a ladder pattern. Specifically, after 2016, the development rate was significantly accelerated, being not only higher than the previous growth rate but also significantly higher than the growth rate of the marine economy. Moreover, as a new economic form, the digital economy is becoming an important driving force to promote the quality improvement, efficiency change and power upgrading of economic development. It is also the commanding point of a new round of global industrial competition and a new driving force to promote the revitalization of the real economy and accelerate the transformation and upgrading. The unsynchronized trend of the development and fluctuation of the marine economy and the digital economy further reflects the low level of coordination between the two and the fact that the marine economy had not yet fully played the role of amplification, superposition, and multiplication. Therefore, it is necessary to accelerate the adjustment of the development policies of the marine economy and the digital economy, and comprehensively promote the in-depth integration and development of the two. Moreover, there was a partial effect on the coordination level of China's marine economy and digital economy, which was mainly reflected in Liaoning, Shandong, and Fujian. Although the level and coordination of its marine economy were relatively high, the level of its digital economy was at medium and low levels. Relying on the advantages of its marine economy, it showed high coordination and formed a partial effect of a high coordination value but a low digital economy level.

*4.2. Spatial Evolution of the Coordination between the Marine Economy and Digital Economy*

To clearly explain the spatial evolution characteristics of the coordination between the marine economy and digital economy in China's coastal provinces, we examined the coordination level of typical years and created maps according to the classification in Table 2. As can be seen from Figure 3, the coordination between China's marine economy and digital economy showed a spatial distribution pattern of "high in the East, and low in the South and North" from 2012 to 2019. The details are as follows:

(1) Severe disorder: Hebei, Guangxi, and Hainan were in a state of severe disorder in 2012. Hebei's disorder was reduced in 2015. Guangxi's disorder was reduced in 2017. Only Hainan was severely disordered in 2019. As shown in Tables 3 and 4, it can be seen that these provinces and cities were in a relatively low position in the quality level of their marine economies and digital economies, indicating that the development of the marine economies and digital economies in these provinces was relatively slow. These areas were strongly constrained by marine resources. The traditional marine traditional industry was dominant, and the scale of the marine economy was not significantly improved. During its development, the marine economy was not deeply integrated with the digital economy represented by big data, 5G, and other digital technologies, and digital technology did not fully empower the marine economy.

(2) Mild disorder: In 2012, Liaoning, Tianjin, and Fujian were in a state of mild disorder. Fujian's disorder was reduced and Hebei was added to this category in 2015. Guangxi was added to this category in 2017. Tianjin and Guangxi were in a state of mild disorder in 2019. This means that the coordination of the marine economy and the digital economy of coastal provinces and cities changed rapidly. The optimal spatial distribution of the marine economy and the transformation and upgrading of the modern marine industrial system promoted the high-quality development of the marine economy. Digital technology was combined with the marine economy to a

certain extent, but the construction of the digital ocean was not comprehensively improved, and the factors of production still had difficulty flowing freely between the marine economy and the digital economy.

(3) Barely coordinated: In 2012, Shandong, Jiangsu, Shanghai, and Zhejiang were in a state of being barely coordinated. Moreover by 2015, Shandong and Jiangsu's disorders were reduced, while Liaoning was added to this category. By 2017, Zhejiang was removed from this category. By 2019, Shanghai's disorder was reduced and Hebei was added to this category. As shown in Tables 3 and 4, we know that with the in-depth implementation of the Belt and Road Initiative and the integrated development strategy of the marine and land economy, the marine economy development quality of China's coastal provinces and cities clearly improved. The digital economy was fully launched, and digital technology gradually enabled the development of the marine economy. The marine industry began a digital transformation, and the relationship between the two was gradually strengthened to promote the transition of coordination.

(4) Primary coordination: In 2012, only Guangdong was in a state of primary coordination. By 2015, Shandong and Jiangsu had been added to this category. By 2017, Zhejiang had been added to this category. By 2019, Shanghai had been added to this category, Guangdong's disorder had been reduced, and the four provinces and cities had entered the state of primary coordination. The provinces and cities that reached the primary coordination state were relatively stable, with relatively small fluctuations and long duration, indicating that the primary coordination stage belonged to the stable run-in period of the development of the marine economy and digital economy. The two formed a relatively fixed interaction mode, but it was difficult to break through into a new state in the near future. It means that these provinces and cities needed to continuously improve the quality of the marine economy and digital economy, rationally plan the marine industry and spatial layout, constantly improve the modern marine industry system, and realize leapfrog development of the marine economy. Moreover, we should accelerate the implementation of the development strategy of the digital economy, focus on the key links, promote the research on the applicable technologies of the characteristic marine industry and the common key technologies in the frontier marine field, and further enhance the coordination between the marine economy and digital economy.

(5) Moderate coordination: In 2019, only Guangdong was in a state of moderate coordination, which indicates that the marine economy and digital economy of Guangdong initially formed a benign interactive development situation. According to the data in Tables 3 and 4, we can see that the level of Guangdong's marine economy and digital economy far exceeded that of other coastal provinces over the 8 years studied. The most important reason was that Guangdong accelerated the transformation and upgrading of the marine industry, fostered new industries, optimized the marine spatial structure, and built a modern marine industrial system. Moreover, it accelerated the promotion of industrial digital transformation and the development of digital industrialization, realized the rapid expansion of the scale of the digital economy, enhanced the competitive advantage of the digital economy, and deepened the integration and coordinated development of the digital economy and the marine industry. However, from the perspective of coordination value, Guangdong was still at the middle end of intermediate coordination, and it still needed to plan the development of the marine economy and digital economy at multiple levels to reach a higher level of coordination.

(6) Senior coordination: At this point, the coordination between China's marine economy and digital economy had not reached the senior coordination state. This was mainly because these provinces were in a period of rapid development, but the extensive development model resulted in the decline of marine resources, the pollution of the offshore environment, and the unhealthy state of the marine ecology. Moreover, the deep integration of the digital economy and the marine economy had not yet been

realized, and the leading role of the digital economy had not yet been fully assumed. With the continuous development of the marine economy and digital economy, the digital transformation of the traditional marine industry and the empowerment of digital technology will be accelerated, and the coordination between the marine economy and digital economy will also be steadily improved, ultimately reaching a state of advanced coordination.

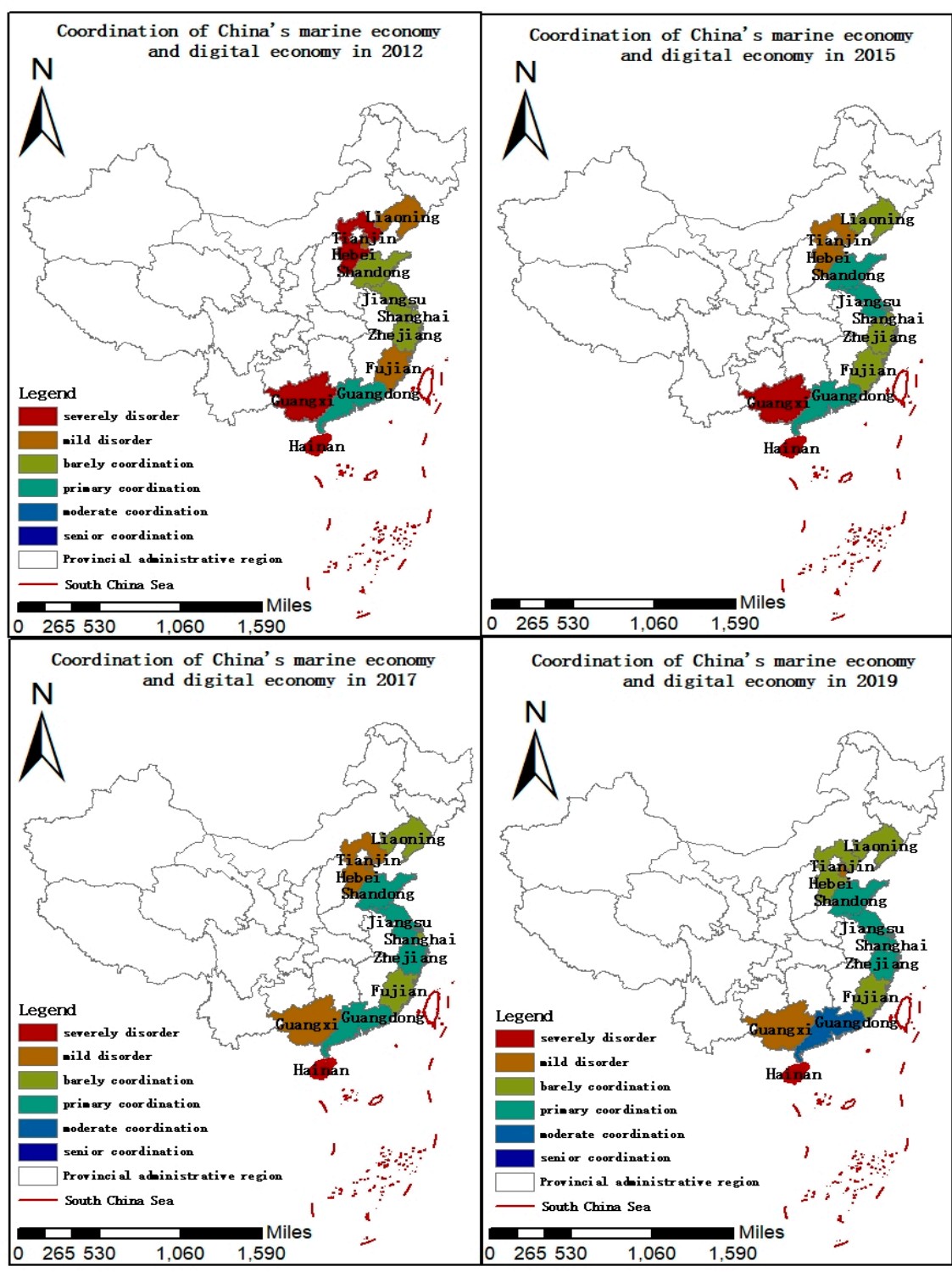

**Figure 3.** Spatial pattern of coordination between the marine economy and digital economy in China in typical years.

*4.3. Analysis of the Interval Difference of Coordination between the Marine Economy and Digital Economy*

To further analyze the coordination spatial differences between the marine economy and digital economy, Theil index was used to calculate the differences among the three marine economy circles. Based on the Theil index model formula, we obtained the detailed results shown in Table 6.

**Table 6.** Theil index of the degree of coordination between the marine economy and digital economy in China (2012–2019).

| Year | Group | | | | | | | |
|---|---|---|---|---|---|---|---|---|
| | Northern Marine Economy Circle | | Eastern Marine Economy Circle | | Southern Marine Economy Circle | | Between-Column | |
| | Theil Index | Contribution Rate (%) | Theil Index | Contribution Rate (%) | Theil Index | Contribution Rate (%) | Theil Index | Contribution Rate (%) |
| 2012 | 0.0045 | 16.8 | 0.0002 | 0.71 | 0.0183 | 67.6 | 0.0041 | 15.1 |
| 2013 | 0.0051 | 19.5 | 0.0003 | 1.21 | 0.0165 | 64.1 | 0.0039 | 15.2 |
| 2014 | 0.0047 | 18.7 | 0.0003 | 1.11 | 0.0162 | 65.3 | 0.0038 | 15.2 |
| 2015 | 0.0044 | 17.9 | 0.0003 | 1.11 | 0.0165 | 66.6 | 0.0036 | 14.4 |
| 2016 | 0.0044 | 17.6 | 0.0004 | 1.46 | 0.0167 | 67.1 | 0.0034 | 13.8 |
| 2017 | 0.0042 | 16.6 | 0.0004 | 1.53 | 0.0175 | 69.3 | 0.0032 | 12.6 |
| 2018 | 0.0048 | 19.1 | 0.0004 | 1.71 | 0.0169 | 67.1 | 0.0031 | 12.1 |
| 2019 | 0.0041 | 16.6 | 0.0005 | 2.06 | 0.0171 | 69.6 | 0.0030 | 11.8 |
| Mean value | 0.0045 | 17.8 | 0.0003 | 1.31 | 0.0169 | 67.1 | 0.0035 | 13.8 |

Source: Obtained by the author.

According to the calculation of the Theil index, the coordination of groups between the marine economy and digital economy was "high in the North and South and low in the East", and the gap between the groups was gradually narrowing from 2012 to 2019. The Theil index of the regional groups followed the order: EMEC, NMEC, SMEC from small to large, with average values of 0.0003, 0.0045, and 0.0169, respectively. The intra-group differences between the NEMEC and the SMEC showed a decreasing trend over the 8-year period, indicating that the differences between the marine economy and digital economy decreased, showing a gradual convergence trend. According to the value of the Theil index, the difference between the coordination of the marine economy and digital economy in the SMEC was still large, with the largest gap found among the three major regions. This indicates that the coordination between the regional marine economy and the digital economy in the SMEC was extremely uneven. The overall strength of the marine economy and digital economy in Guangdong was relatively high, with a large gap between Hainan and Guangxi, and there was a high degree of primacy in the region, forming a polarization effect. Therefore, the SMEC should deepen the promotion of integrated development, break the restrictions of regional administrative barriers on the flow of factors, and improve the mechanism of the integrated development system. Taking the Guangdong–Hong Kong–Macao Greater Bay Area as the center, we should strengthen exchanges and cooperation in market, talent, capital, and technology; drive the integrated development of the digital economy and marine economy in other provinces and cities in the region; and improve the coordination level of the regional marine economy and digital economy. The gap between the coordination of the marine economy and digital economy in the NMEC was still evident, although the difference was narrowing. Therefore, we should accelerate the development of integration; strengthen the free flow of factors between provinces; accelerate the integration of industrial innovation, the construction of market factors on the basis of the community of destiny, the community of interests, and the community of responsibility; and further improve regional coordination and balance. The Theil index of the EMEC was the lowest, indicating that the coordinated development within the region

was relatively balanced. However, the gap within the region was gradually widening, and considering the coordination level of the regional marine economy and digital economy, the overall coordination level of the marine economy and digital economy was not high, only reaching the primary coordination level. The lower coordination level of the regional marine economy and digital economy was directly related to the development level of the regional marine economy and digital economy. Therefore, it is important to accelerate the improvement of the development quality of the regional marine economy and digital economy, enhance the advantages of the digital economy, raise the energy level of the marine economy, and promote the deep integration of the marine economy and digital economy.

According to the change in the Theil index among regions, the interregional index had a large decline, with an average annual decline of 2.1%, and the contribution rate decreased from 15.1% to 11.8%. This was primarily because the in-depth implementation of the strategies of digital economy development and regional integration development set up a cross-regional cooperation platform for coastal provinces and cities, which stimulated the vitality of the regional marine economy, fostered new growth drivers, and expanded new industries, injecting new vitality into the high-quality development of the marine economy. Moreover, China accelerated the development of the digital economy, promoted integration with the real economy, enabled the upgrading of traditional industries, continuously improved the overall efficiency of economic development, and strengthened the resilience of the regional economy, making the coordinated development of the regional marine economy and digital economy change from low quality, disordered, and imbalanced to high quality, sustainable, and balanced.

## 5. Factors That Affected Coordination between Marine Economy and Digital Economy

### 5.1. Model Specification

As we all know that, the coordination between the marine economy and digital economy is affected by many factors and accurately identifying the various factors is important to improve the coordination between them. Based on the available literature, we selected the marine ecological governance (Meg), marine industrial level (Mil), marine economy scale (Mes), marine innovation capability (Mst), digital governance (Dgl), digitization of industry (Dml), digital industrialization (Des), and digital infrastructure level (Dci) as the main influencing factors. We also took the nearshore and coastal area, the output value of the marine tertiary industry, the volume of marine cargo, the number of marine patents authorized, the investment in fixed assets of information software and information technology service industry, the sales volume of e-commerce, the volume of telecommunication business, and the number of Internet broadband access users as the specific indicators. The following random effects Tobit model was established:

$$
\begin{aligned}
Cor = {} & \beta_0 + \beta_1 Ln(Meg) + \beta_2 Ln(Mil) + \beta_3 Ln(Mes) + \beta_4 Ln(Mst) + \\
& \beta_5 Ln(Dgl) + \beta_6 Ln(Dml) + \beta_7 Ln(Des) + \beta_8 Ln(Dci) + \varepsilon
\end{aligned}
\tag{11}
$$

In the model, *Cor* is the dependent variable, which represents the coordination between the marine economy and digital economy. The value has the range [0, 1]; $\beta_i$, $i = (0, 1, 2, \cdots, 8)$, are the undetermined coefficients; and $\varepsilon$ is the random error term.

### 5.2. Descriptive Statistics

As can be seen from Table 7, the average coordination between the marine economy and digital economy from 2012 to 2019 was 0.331, showing that the general coordination between the marine economy and digital economy was low. The whole standard deviation of all indexes was small, which indicates that the sample statistics and the overall parameter values were relatively similar, and the sample was representative.

**Table 7.** Descriptive statistics of the variables.

| Variables | Symbol | Observation | Mean Value | Standard Deviation | Minimum Value | Maximum Value |
|---|---|---|---|---|---|---|
| Coordination level | Cor | 88 | 0.3311 | 0.1197 | 0.1079 | 0.6787 |
| Marine ecological governance | Meg | 88 | 3.7491 | 0.7167 | 1.7596 | 4.6895 |
| Marine industry level | Mil | 88 | 2.8831 | 0.3686 | 0.5271 | 3.4082 |
| Marine economy scale | Mes | 88 | 2.7881 | 0.8971 | 0.9517 | 4.3741 |
| Marine innovation capacity | Mst | 88 | 4.5015 | 1.6418 | 0.6931 | 7.2196 |
| Digital governance | Dgl | 88 | 2.9311 | 0.8141 | 0.4511 | 4.7357 |
| Digitalization of industry | Dml | 88 | 3.4891 | 1.1635 | 0.4447 | 5.7094 |
| Digital industrialization | Des | 88 | 2.3771 | 1.0255 | 0.0759 | 4.8967 |
| Digital infrastructure level | Dci | 88 | 4.5938 | 0.8992 | 2.2569 | 5.9406 |

*5.3. Empirical Result Analysis*

As can be seen from Table 8, model 4 had the best calculation effect. The empirical results show that the degree of influence of each factor on the coordination of the marine economy and digital economy was mainly affected by the scale of the marine industry, digital infrastructure, marine industry level, industrial digitalization, marine innovation capacity, digital industrialization, digital governance, and marine ecological governance. Among them, the influence coefficients of the marine economy scale, the marine industry level, and the marine innovation capacity were 0.063, 0.021, and 0.012, respectively, and all passed the significance test at 1%. Among them, the influence coefficient of the marine economy scale was the highest, reaching 0.063 and passing the 1% significance test, which means that for every 0.01 increase, the coordination between the marine economy and digital economy increased by 0.063. This was limited by the fact that as the scale of the marine economy increased, the industries, markets, and other fields that it affected naturally expanded, and the interaction and coupling coordination with the digital economy were also improved. The level of the marine industry also had a positive impact on the coordination of the two. Improvement in the level of the marine industry level can strengthen the marine industrial chain and supply chain, drive the development of upstream and downstream industries, and inevitably strengthen the connection with the digital economy and improve its coordination. The capability of marine science and technology innovation also had a positive impact on the coordination between the marine economy and digital economy. Also the enhancement of scientific ability and technological innovation can improve the capability of scientific and technological transformation and the spillover effect, and it can strengthen integrated development with the digital economy, thus promoting the improvement of the coordination between the marine economy and digital economy.

Digital governance, industrial digitalization, digital industrialization, and digital infrastructure all had positive impacts on the coordination of the marine economy and digital economy, and the impact coefficients were 0.0094, 0.0134, 0.0101, and 0.0316, respectively. Digital governance passed the significance test at 10%, and the rest passed the significance test at 5%. Among them, the impact coefficient of digital infrastructure on the coordination between the marine economy and digital economy was the highest, and the coordination increased by 0.0316 percentage points for each percentage point increase. This fully indicates that digital infrastructure was an important bridge for promoting the coordination between the marine economy and digital economy, and it could promote the effective flow of data elements between the marine economy and digital economy, forming a benign interaction process. Digital governance, industrial digitalization, and digital industrialization had similar effects on the coordination between the marine economy and digital economy. Each increase by one percentage point increased the coordination by about 0.01 percentage points, which indicates that the development of industrial digitalization

and digital industrialization could promote the deep integration of the digital economy and marine industry. This could promote transformation and upgrading of the marine industry and cultivate new forms of the marine industry. Moreover, the enhancement of digital governance could promote the digital level of marine governance and strengthen the construction and management of the digital ocean.

**Table 8.** Model regression results.

| Variables | Model 1 Fixed-Effects OLS | Model 2 Random-Effects OLS | Model 3 Hybrid Model Tobit | Model 4 Random-Effects Tobit |
|---|---|---|---|---|
| Marine ecological governance (Meg) | 0.0165 (0.0231) | 0.00883 (0.0155) | 0.0203 (0.0187) | 0.00781 (0.0120) |
| Marine industry level (Mil) | 0.0201 * (0.00922) | 0.0196 ** (0.00630) | −0.0185 (0.0209) | 0.0212 *** (0.00645) |
| Marine economy scale (Mes) | 0.0829 ** (0.0227) | 0.0492 ** (0.0170) | 0.0226 ** (0.00953) | 0.0628 *** (0.0115) |
| Marine innovation capacity (Mst) | 0.0130 * (0.00699) | 0.0124 ** (0.0057) | 0.0118 * (0.00625) | 0.0123 *** (0.00315) |
| Digital governance (Dgl) | 0.00825 * (0.00434) | 0.0103 ** (0.00436) | 0.0252 ** (0.00808) | 0.00942 * (0.00481) |
| Digitalization of industry (Dml) | 0.0108 (0.00835) | 0.0171 ** (0.00599) | 0.0396 *** (0.0106) | 0.0134 ** (0.00631) |
| Digital industrialization (Des) | 0.00975 (0.00612) | 0.00963 * (0.00566) | 0.0176 ** (0.00841) | 0.0101 ** (0.00512) |
| Digital infrastructure (Dci) | 0.0278 ** (0.0115) | 0.0338 ** (0.0113) | 0.00556 (0.0170) | 0.0316 ** (0.0129) |
| Constant term | −0.291 ** (0.0813) | −0.219 *** (0.0527) | −0.0872 ** (0.0302) | −0.233 *** (0.0556) |
| var(e.y) | | | 0.000777 ** (0.000259) | |
| sigma_u | | | | 0.0360 *** (0.00921) |
| sigma_e | | | | 0.0144 *** (0.00120) |
| N | 88 | 88 | 88 | 88 |

Note: *, **, and *** indicate significance at the 10%, 5%, and 1% levels, respectively.

## 6. Conclusions and Countermeasures

Based on the calculation of the level, Theil index, and coordination of marine economy and digital economy in coastal provinces and cities of China from 2012 to 2019, the following key observations were made.

### 6.1. Conclusions

First, the marine economy and digital economy development level in China's coastal provinces and cities showed an overall upward with time, but the overall level was low. In terms of spatial evolution, the development levels of the marine economy and digital economy both presented a spatial distribution pattern of "high in the East and low in the South and North". The gap in the development level of the digital economy between provinces gradually widened.

Second, the coordination between the two of China's coastal provinces and cities steadily increased, but the absolute value was small. The average coordination between the marine economy and digital economy in Guangdong was the highest, while that between Guangxi and Hainan was the lowest, and the average coordination between other provinces and cities was between 0.3 and 0.4, with a small difference. There were partial effects in the coordinated evolution of the regional marine economy and digital economy. From the perspective of spatial evolution, the coordinated evolution of the marine economy and digital economy showed a spatial pattern of "high in the East and low in the South and North". The coordination between the marine economy and digital economy was in the

dynamic adjustment period, most provinces and cities were in the barely coordinated and primary coordination stages, and only Guangdong was in the intermediate coordination stage in 2019.

Third, the coordination gap between the regional marine economy and digital economy was obvious, showing a spatial distribution pattern of a "high gap in the South and low gap in the East", but the gap within and between groups gradually narrowed. The Theil index in the regional groups was divided into the East Marine Economy Circle, the North Marine Economy Circle, and the South Marine Economy Circle from small to large. Also the interregional index dropped significantly from 15.1% to 11.8%, with an average annual decline of 2.1%. The coordinated development of the regional marine economy and digital economy transformed from low-quality, disordered, and unbalanced to high-quality, sustainable, and balanced.

Fourth, through the model calculation, it was identified that the main factors that influenced the coordination between the marine economy and digital economy in China's coastal provinces and cities were digital infrastructure construction level, the marine economy scale, the marine industry level, industrial digitization, and the innovative capacity of marine science and technology, all of which had a positive impact on the coordination between the two.

*6.2. Countermeasures and Suggestions*

To comprehensively improve the coordination level between the marine economy and digital economy, promote high-quality development of the marine economy and digital economy, and create new competitive advantages in the new development stage, the following actions should be taken.

First, improve the scale of the marine economy. Focus on building upstream and downstream industrial clusters of marine data centers, intelligent terminal industrial clusters, and digital integration industrial clusters, and promote the extension of the marine industrial chain to the middle and high ends. Actively develop smart marine tourism, smart marine information services, and other new business forms; promote smart port shipping logistics; build a comprehensive logistics park of marine informatization; form a smart park for the marine industry; and accelerate the expansion of the marine economy scale.

Second, improve the construction of digital infrastructure. Accelerate the layout of digital infrastructure, build new types of facilities (e.g., artificial intelligence, cloud computing, and big data), transform and upgrade broadband network facilities, and integrate new infrastructure with traditional infrastructure. Actively promote the connectivity between new digital facilities and marine network facilities, and utilize new generation information technologies, such as 5G, big data, artificial intelligence, and the Internet of things, to build a marine big data center and marine information service platform that integrates marine environmental target perception, transmission, data processing, and information services. This way, we can realize the integration of land and marine facilities. Build a 3D marine information network; improve the "ship network", "shore network", "intelligent shipping network", and intelligent marine platform; and strengthen the marine digital service guarantee.

Third, accelerate the marine industry digital development. Promote the "cloud empowerment and intelligence" of the traditional marine industry; make full use of digital technologies (e.g., big data, Internet of things, artificial intelligence, and blockchain) to transform the traditional marine industry; build smart fishing ports, smart marine ranches, smart marine manufacturing, and smart ports; give full play to the role of digital technologies in amplifying and doubling the marine industry; and raise the level of digitalization of the marine industry.

Fourth, promote the innovative development of marine science and technology. Carry out systematic and chain design for scientific and technological innovation in the marine industry; promote marine enterprises to improve the innovation system; promote the deep integration of the industrial chain, talent chain, capital chain, and innovation chain; and

realize the optimal allocation of innovation elements. We will support and guide industry leaders and specialized, innovative enterprises with key core technologies to further reform and strengthen innovation.

## 7. Discussion

At present, the world economy has entered the digital era, and the relationship between the marine economy and digital economy has become increasingly close. In 2020, the Digital Twin of the Ocean (DTO) project, launched by the European Union (EU) and the United Nations (UN), proposed that the DTO is helping us to deepen our understanding of the ocean, improve our ability to continuously monitor the ocean, and enable us to predict the evolution of the ocean and manage marine resources in a sustainable way [56]. In 2021, the United Nations Conference on Trade and Development issued the "Digital Economy Report (Data Cross-border Flow and Development: Data Flow for Who)", which pointed out that digital data, as an economic and strategic resource, play an increasingly important role, and that the governance of data and data flow plays an important role in solving the problems of information asymmetry, data ethics, data filtering, and review in the marine economy [57]. Moreover, the UN announced that 2021 was the beginning of the "Decade of the Ocean". One of the 10 challenges of the plan is to create an integrated digital virtual body of the ocean. It can be seen that comprehensively strengthening the in-depth integration and development of the marine economy and the digital economy has become a major issue facing all countries. Therefore, it is not only a scientific issue but also a practical issue to explore how to promote the coordinated development of the marine economy and the digital economy from multiple perspectives and achieve a stronger synergy between the two. This is of great significance for promoting the sustainable development of the marine economy and the high-quality development of the digital economy.

Studying the coordination between China's marine economy and digital economy has important theoretical and practical value for the high-quality development of the marine economy of other coastal countries and regions around the world, which is mainly reflected in the following two aspects.

First, it provides a new perspective for other countries and regions to study the marine economy.

At present, researchers' contributions regarding the marine economy and digital economy mostly focused on a single object, and few scholars used them as a system for coupling and coordination analysis. The marine economy and digital economy are two complex and open systems, and there are multiple tight links between them. A one-sided emphasis on any system will lead to research deviation, which can not only hinder the sustainable development of the marine economy and digital economy but also cause resource redundancy and development lag. The coupling coordination model was commonly used to measure the coordination level of the two, reveal the coupled dynamic evolution state and laws, analyze the regional differences of their coordination, and identify the factors that affected their coordination. This study can provide a new research perspective for the world's coastal countries and regions to study the coordination of the marine economy and digital economy, change the traditional concept of dividing the development of the marine economy and digital economy, and enrich their research content.

Second, the study provides a reference for other countries and regions to formulate digital development policies for the marine economy. According to the model calculation, the main factors that affected the coordination of the marine economy and digital economy included the level of infrastructure construction, the scale of the marine economy, the level of the marine industry, digitalization of the industry, and the innovation ability of the marine science and technology. Therefore, when we improve the coordination between the marine economy and digital economy, we should focus on improving these policies and on giving full play to their synergy and amplification. These research results can also provide an important reference for coastal countries and regions around the world to improve the digital development policies of the marine economy. Furthermore,

different countries and regions have different development characteristics and objectives, and corresponding countermeasures and measures should also be taken according to the actual local development situation, highlighting the key points and adjusting measures to local conditions.

This study mainly measured and analyzed the coordination between the marine economy and digital economy in coastal provinces and cities of China. However, owing to the availability of data, the coordination between the marine economy and digital economy in coastal cities was not thoroughly analyzed. Moreover, the analysis of the spatial spillover effect of the coordination between the marine economy and digital economy was relatively limited in this study, which is a shortcoming of this study, and this will be the focus of future research. Additionally, other work can consider adopting qualitative or statistical methods to analyze the coordination between the marine economy and digital economy. It will also be valuable to study the corresponding policy measures and governance strategies.

**Author Contributions:** Conceptualization, data analysis, and original draft, Y.L.; methodology, software, and visualization, Y.J. and Z.P.; writing—review and editing, N.X.; supervision, project administration, and funding acquisition, A.W. All authors have read and agreed to the published version of the manuscript.

**Funding:** Social Science Planning Fund of Liaoning's "Research on high-quality development path of marine economy in Liaoning" (L22AJL002). Project approved by Liaoning Provincial Department of Education's "Research on the digital transformation and development of Liaoning marine industry" (KJKMR20221130).

**Institutional Review Board Statement:** Not applicable.

**Informed Consent Statement:** Informed consent was obtained from all subjects involved in the study.

**Data Availability Statement:** The data presented in this study are available on request from the corresponding author.

**Acknowledgments:** The authors would like to thank Antonella Petrillo for this kind and insightful advice. We thank two anonymous reviewers for constructive comments that improved this manuscript.

**Conflicts of Interest:** The authors declare no conflict of interest.

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
