# Peer review of "Evolution of the Coupling Coordination between the Marine Economy and Digital Economy"

_sustainability, doi:10.3390/su15065600_

Round 1
Reviewer 1 Report
The article is interesting. The authors address a new aspect of the development of the global and regional economy, revealing territorial differentiation. The course of the study and the results are of interest to a wide range of readers.
The article is well written, but it has defects:
1) The authors argue that the maritime economy and the digital economy are developing in waves. But they do not show whether these waves were synchronous or asynchronous. There is no evidence or discussion of this. The presence or absence of shifts affects the decisions that can be made for the coordinated development of the industries.
2) The authors look at the digital economy as a factor in the development of the marine economy. But the authors barely touch on the aspect of the marine economy's impact on the digital economy. Lines 127-139 provide scant evidence of this.
3) The authors need to define more clearly the composition and structure of the marine economy and the digital economy. For this purpose, the national classifier of economic activities can be used to indicate which industries make up these activities.
4) The authors did not justify the choice of research methods (section 2.3). Why did the authors take Entropy method, Coupling harmonious degree model, Theil index, Tobit model? There is no comparison with other methods that can be used to solve this problem. A detailed review is needed.
5) The authors did not specify the reasons for the territorial differentiation of the marine economy and the digital economy of the provinces (section 3). Determining the level of development is not the final task. The authors should show how these values can be used in decision making.
The article can be reviewed again after substantial corrections.
Author Response
Dear Reviewers:
Thank you for your letter concerning our manuscript entitled “Evolution of the coupling coordination between marine economy and digital economy” (sustainability: 2205818). Those comments are all valuable and very helpful for revising and improving our paper, as well as the important guiding significance to our researches. We have studied comments carefully and have made correction that we hope meet with approval. Revised portion are marked in red in the paper. (Please refer to the revised version for details.) The main corrections in the paper and the responds to the reviewer’s comments are as following. (Please refer to the page number of the revised version of the manuscript without comments.)
Responds to the reviewer 1’s comments:
The article is interesting. The authors address a new aspect of the development of the global and regional economy, revealing territorial differentiation. The course of the study and the results are of interest to a wide range of readers. The article is well written, but it has defects:
- The authors argue that the maritime economy and the digital economy are developing in waves. But they do not show whether these waves were synchronous or asynchronous. There is no evidence or discussion of this. The presence or absence of shifts affects the decisions that can be made for the coordinated development of the industries.
Response: Thanks very much for the guidance of the Reviewer. Possibly we may not express clearly in the manuscript, and cause misunderstanding. The comments from reviewers are of great reference significance. The revisions are as following:
(1) We have already discussed the changing trend of the marine economy and digital economy. The development and fluctuation trends of the marine economy and digital economy in China’s coastal provinces and cities were not synchronized. (Part 4.1, P14)
According to Table 3, 4 and 5, the development and fluctuation trends of the marine economy and digital economy in China's coastal provinces and cities were not synchronized. During the study period, the marine economy developed slowly and fluctuated greatly, and some provinces and cities even experienced a downward trend. This was mainly due to the fact that China's marine economy was in the period of optimizing and adjusting the industrial structure and accelerating the conversion of new and old kinetic energy, and the marine industry modernization system was being established. However, the development of the digital economy has a small fluctuation, and the overall development trend was in a ladder pattern. Especially after 2016, the development speed has been significantly accelerated, which was not only higher than the previous growth rate, but also significantly higher than the growth rate of the marine economy. As a new economic form, the digital economy has developed rapidly in China, and has gradually become an important force in restructuring factor resources, reshaping the economic structure and changing the competition pattern. The unsynchronized trend of the development and fluctuation of the marine economy and the digital economy further reflected the low level of coordination between the two and has not yet fully played the role of amplification, superposition and multiplication. It is urgent to accelerate the adjustment of the development policies of the marine economy and the digital economy, and comprehensively promote the in-depth integration and development of the two.
- The authors look at the digital economy as a factor in the development of the marine economy. But the authors barely touch on the aspect of the marine economy’s impact on the digital economy. Lines 127-139 provide scant evidence of this.
Response: Thanks very much for the guidance of the Reviewer. The comments from reviewers are of great reference significance. The revisions are as following:
(1) We systematically discussed the relationship between marine economy and digital economy. We also supplemented some statements about the impact of digital economy. (Part 2.1 of P4)
Second, the digital economy provides new impetus for the development of the marine economy. The digital economy can empower the marine economy, promote the optimization and upgrading of the marine industrial structure, glow new vitality, form new growth points, and improve the resilience and development quality of the marine economy. The digital economy uses data elements to lead the free flow and reorganization of technology, capital and material flows of the marine economy, and build a new modern industrial system. At the same time, the digital economy itself has also spawned more new industries and new formats to promote the upgrading and expansion of the marine economy. The digital economy can accurately identify the real needs of the marine economy industry chain, effectively avoid excessive energy consumption, environmental pollution and other problems in the development of the marine industry, and promote the intelligent development of the marine economy. Digital industrialization can lead the optimization and reorganization of marine economic factors with data flow, break the space-time constraints of the flow of production factors, realize the free flow of marine economic production factors, promote innovative products and services in the field of marine economy, and reshape the pattern of marine economy.
- The authors need to define more clearly the composition and structure of the marine economy and the digital economy. For this purpose, the national classifier of economic activities can be used to indicate which industries make up these activities.
Response: Thanks very much for the guidance of the Reviewer. The comments from reviewers are of great reference significance. Due to the limitation of the layout, we have not discussed the selection and composition of indicators of marine economy and digital economy in details. The reasons are explained as follows:
(1) The marine economy contains a lot of contents, and this paper mainly uses the conventional division method. So we chose ecological environment, industrial level and economic scale as the three indicators. The marine economy development emphasizes green and sustainable development, so innovation capability is selected as an important indicator to measure the marine economy. The selection is mainly to measure the comprehensive strength and quality of the marine economy, not just the industrial part of the marine economy.
(2) With regard to the digital economy, the definition of digital economy in China is mainly elaborated from four aspects: digital governance, industry digitization, digital industrialization and digital infrastructure, and the existing literature also adopts this classification method.
- The authors did not justify the choice of research methods (section 2.3). Why did the authors take Entropy method, Coupling harmonious degree model, Theil index, Tobit model? There is no comparison with other methods that can be used to solve this problem. A detailed review is needed.
Response: Thanks very much for the guidance of the Reviewer. The comments from reviewers are of great reference significance. We added the relevant reasons and detailed reviews for the selection of research methods. The specific reasons are as follows:
(1) We have supplemented the literature review of methods in the part of entropy method, coupling harmonious degree model, Theil index, Tobit model.(Part 2.3.1, 2.3.2, 2.3.3,2.3.4, P8-10) The literature reviews are as follows:
[50] Zadeh, L. (1965). A. Fuzzy sets. Information and control, (03), 338-353.
[51] Burillo, P., & Bustince, H. (1996). Entropy on intuitionistic fuzzy sets and on interval-valued fuzzy sets [J]. Fuzzy Sets Systems, (03), 305-316.
[52] Qijiao Song, Nan zhou,et.al. Investigation of a “coupling model” of coordination between low-carbon development and urbanization in China[J]. Energy Policy,121(2018):346-354. https://doi.org/10.1016/j.enpol.2018.05.037.
[53] O.V. Rayevneva, O.Y. Bobkova, Identifying sources of development disparities of Ukraine’s regions basing on decomposition of Theil index [M]. Actual Probl. Econ. 128 (2012) 200–210.
[54] J. Bhattacharya, A. Sinha, Inequality in per capita water availability: a Thiel’s
second measure approach, Desalin[J].Water Treat. 57 (1) (2016) 136–144.
[55] Takeshi Amemiya. Tobit models: A survey [J].Journal of Econometrics.1984(24):
3-61.https://doi.org/10.1016/0304-4076(84)90074-5.
- The authors did not specify the reasons for the territorial differentiation of the marine economy and the digital economy of the provinces (section 3). Determining the level of development is not the final task. The authors should show how these values can be used in decision making.
Response: Thanks very much for the guidance of the Reviewer. The comments from reviewers are of great reference significance. Due to the limitation of the layout, we have not discussed the selection and composition of indicators of marine economy and digital economy in details. The reasons are explained as follows:
This paper has made some contributions in the following aspects: First, from the theoretical level, it discusses the relationship between the development of marine economy and digital economy, and enriches the literature of sustainable development of marine economy. Secondly, from an empirical perspective, it analyzes the characteristics of the coordinated space-time evolution and regional differences of marine economy and digital economy, and reveals the dynamic adjustment process of their coordination. Thirdly, taking the major coastal cities in China as an example, this paper discusses the coordinated countermeasures and suggestions for improving the marine economy and digital economy from the aspects of industrial coordination, infrastructure connectivity, and ecological coordinated governance.
In addition, we proofread the full text and some spelling mistakes carefully. Thank you for your patient guidance and careful reading. Also, we have entrusted professional institutions to proofread and polish the language of the manuscript to improve the accuracy of the article.

Reviewer 2 Report
The reviewed work is interesting. However, it contains a lot of repetition. It is poorly rooted in the non-western literature. The authors seem to have a one-sided understanding of the digital economy, too narrowly.
In addition, which is very important from a content point of view. Figure 3 is cartographically poorly done. The cartogram method depicts the intensity of a phenomenon by means of shades of one color or the density of one type of gray. Now this is a surface method, which is not used to depict numerical data.
Author Response
Dear Reviewers:
Thank you for your letter concerning our manuscript entitled “Evolution of the coupling coordination between marine economy and digital economy” (sustainability: 2205818).Those comments are all valuable and very helpful for revising and improving our paper, as well as the important guiding significance to our researches. We have studied comments carefully and have made correction that we hope meet with approval. Revised portion are marked in red in the paper. (Please refer to the revised version for details.) The main corrections in the paper and the responds to the reviewer’s comments are as following. (Please refer to the page number of the revised version of the manuscript without comments.)
Responds to the reviewer 2’s comments:
- The reviewed work is interesting. However, it contains a lot of repetition. It is poorly rooted in the non-western literature. The authors seem to have a one-sided understanding of the digital economy, too narrowly.
Response: Thanks very much for the guidance of the Reviewer. Possibly we may not express clearly in the manuscript, and cause misunderstanding. The comments from reviewers are of great reference significance. We proofread the full text and some spelling mistakes carefully. Thank you for your patient guidance and careful reading. Also, we have entrusted professional institutions to proofread and polish the language of the manuscript to improve the accuracy of the article.
- In addition, which is very important from a content point of view. Figure 3 is cartographically poorly done. The cartogram method depicts the intensity of a phenomenon by means of shades of one color or the density of one type of gray. Now this is a surface method, which is not used to depict numerical data.
Response: We are grateful for the suggestion and we highly appreciate your time and consideration. To be more clearly and in accordance with the reviewer concerns, we made some explanations as follows:
The purpose of using GIS map in this manuscript is to describe the coordination status of marine economy and digital economy in coastal areas more clearly and intuitively. The GIS map reflects the time evolution, characteristics and differences of the coordination level between marine economy and digital economy through color changes, rather than describing numerical data.

Reviewer 3 Report
Elaborate on the key contributions of the paper.
Please elaborate on the theoretical underpinnings or models backing your topic. Currently the theoretical section is more of a conceptual review.
Literature review is totally missing. Please include that.
You need to elaborate on the overall methodology and data specifics.
You need to align your results with prior literature.
Inline 712- do your mean lack of data ...
Author Response
Dear Reviewers:
Thank you for your letter concerning our manuscript entitled “Evolution of the coupling coordination between marine economy and digital economy” (sustainability: 2205818).Those comments are all valuable and very helpful for revising and improving our paper, as well as the important guiding significance to our researches. We have studied comments carefully and have made correction that we hope meet with approval. Revised portion are marked in red in the paper. (Please refer to the revised version for details.) The main corrections in the paper and the responds to the reviewer’s comments are as following. (Please refer to the page number of the revised version of the manuscript without comments.)
Responds to the reviewer 3’s comments:
- Elaborate on the key contributions of the paper.
Response: Thanks very much for the guidance of the Reviewer. The comments from reviewers are of great reference significance. Due to the limitation of the layout, we didn’t add the contributions in the manuscript. The key contributions of the manuscript are as follows:
This paper has made some contributions in the following aspects: First, from the theoretical level, it discusses the relationship between the development of marine economy and digital economy, and enriches the literature of sustainable development of marine economy. Secondly, from an empirical perspective, it analyzes the characteristics of the coordinated space-time evolution and regional differences of marine economy and digital economy, and reveals the dynamic adjustment process of their coordination. Thirdly, taking the major coastal cities in China as an example, this paper discusses the coordinated countermeasures and suggestions for improving the marine economy and digital economy from the aspects of industrial coordination, infrastructure connectivity, and ecological coordinated governance.
- Please elaborate on the theoretical underpinnings or models backing your topic. Currently the theoretical section is more of a conceptual review.
Response: We are grateful for the suggestion and we highly appreciate your time and consideration. To be more clearly and in accordance with the reviewer concerns, we made some explanations as follows: (Part 2.1, P3-4)
Marine economy and digital economy are two subsystems of social economic system. From the perspective of system theory, the two have the characteristics of openness and organicity. They are not zero-sum games and contradict each other, but are mutually coupled and mutually beneficial. With the development and penetration of digital technology, the relationship between them is more close, complex and diversified. One-sided emphasis on the digital economy or the marine economy will lead to resource surplus and development lag. Only by realizing complementary advantages and dynamic optimization can the two promote the development from low to high level, and realize the situation of "mutual benefit, symbiosis and win-win cooperation". The coordination relationship between marine economy and digital economy is shown in Figure 1.
First, the marine economy provides the material basis for the development of the digital economy. The marine economy has produced a large amount of data in the process of ecological governance, industrial development, scientific and technological innovation and scale expansion. These data are important data elements for the development of the digital economy and the basis for the development of the digital economy. The sustainable development of the marine economy can create more new application scenarios for the development of the digital economy, provide important application objects for the innovative development of digital technology, and provide broad space for the sustainable development of the digital economy, such as market, talent, resources and information, and become the spatial carrier of the development of the digital economy. The sustained development of the marine economy will generate high-end and diversified demand, which will promote the development of technological innovation in the digital economy and promote the improvement of the innovation capacity of the digital economy.
Second, the digital economy provides new impetus for the development of the marine economy. The digital economy can empower the marine economy, promote the optimization and upgrading of the marine industrial structure, glow new vitality, form new growth points, and improve the resilience and development quality of the marine economy. The digital economy uses data elements to lead the free flow and reorganization of technology, capital and material flows of the marine economy, and build a new modern industrial system. At the same time, the digital economy itself has also spawned more new industries and new formats to promote the upgrading and expansion of the marine economy. The digital economy can accurately identify the real needs of the marine economy industry chain, effectively avoid excessive energy consumption, environmental pollution and other problems in the development of the marine industry, and promote the intelligent development of the marine economy. Digital industrialization can lead the optimization and reorganization of marine economic factors with data flow, break the space-time constraints of the flow of production factors, realize the free flow of marine economic production factors, promote innovative products and services in the field of marine economy, and reshape the pattern of marine economy.
Third, the marine economy and the digital economy interact and co-exist. The marine economy and the digital economy realize deep integration through information, technology, data, talent, market and capital and other factors. The various production factors interact with each other, reorganize the factor resources in dynamic adjustment, optimize the production, circulation, consumption and distribution of the marine economy and the digital economy, enhance the development momentum, improve the quality and efficiency of development, realize the cooperation and symbiosis between the two, and continue to move towards a higher level of coordination.
- Literature review is totally missing. Please include that.
Response: Thanks very much for the guidance of the Reviewer. Possibly we may not express clearly in the manuscript, and cause misunderstanding. The comments from reviewers are of great reference significance. Due to the limitation of the length of the paper, we have omitted some literature reviews. Now it is supplemented, and the supplementary content is in the introduction part. The revisions are as following:
(1) We have added some literature review about marine economy and digital economy in the introduction part.(P2)
Previous research of marine economy has focused on the contribution of the marine economy to the national economy [5-7], on the contributions of the marine element [8,9], and on the economic differences between inland and coastal regions.[10,11] Digital economy refers to an economy that is based on digital technologies. The qualitative research literature argues that digital economy is multi-disciplinary.[12] The driving source of digitalisation essentially is the effective application of information and communications technology (ICT), and digital economy is a new economic form that has emerged with the rapid development of the ICT industry. In addition, the quantitative research literature is concerned about the appropriateness of traditional research approaches and methodologies that attempt to measure the development of the digital economy from different aspects. [13-18]
The literature reviews are as following:
[5] J.T. Kildow, A. Milligan, The importance of estimating the contribution of the oceans to national economies, Mar. Policy 34 (2010) 367–374.
[6] K. Morrissey, C. O’ Donoghue, S. Hynes, Quantifying the value of multi-sectoral marine commercial activity in Ireland, Mar. Policy 35 (2011) 721–727.
[7] K. Morrissey, C. O’ Donoghue, The role of the marine sector in the Irish national economy: an input-output analysis, Mar. Policy 37 (2013) 230–238.
[8] R. Zhao, S. Hynes, G.S. He, Defining and quantifying China’s ocean economy, Mar. Policy 43 (2014) 164–173.
[9] J. Ding, X.Q. Ge, R. Casey, “Blue competition” in China: current situation and challenges, Mar. Policy 44 (2014) 351–359.
[10] T.A. Stojanovic, C.J.Q. Farmer, The development of world oceans and coasts and concepts of sustainability, Mar. Policy 42 (2013) 157–165.
[11] Karyn Morrissey, Using secondary data to examine economic trends in a subset of sectors in the English marine economy:2003−2011, Mar. Policy 50 (2014) 135–141.
[12] Anthony J. Qnwuegbuzie, et.al. Qualitative Analysis Techniques for the Review of the Literature[R]. The Qualitative Report,2012, Volume17, Article 56,1-28. http://www.nova.edu/ssss/QR/QR17/onwuegbuzie.pdf.
[13] Beomsoo Kim, Anitesh Barua, et.al. Virtual field experiments for a digital economy: a new research methodology for exploring an information economy[J].Decision Support Systems, Volume32,2002(1):215-231.
[14] L. Cui et al. Text mining to explore the influencing factors of sharing economy driven digital platforms to promote social and economic development[J]. Information Technology for Development,2020(9):779-801.https://doi.org/10.1080/02681102.2020.1815636.
[15] D.Curran. Risk, innovation, and democracy in the digital economy[J]. European Journal of Social Theory,2018, Vol,2(2):207-26. https://doi.org/10.1177/1368431017710907.
[16] Isabella Jesemann. Support of startup innovation towards development of new industries[J]. Procedia CIRP, Volume88, 2020:3-8. https://doi.org/10.1016/j.procir.2020.05.001.
[17] Loren Brandt, Eric Thun. Constructing a Ladder for Growth: Policy, Markets, and Industrial Upgrading in China[J]. World Development, 2016(4):78-95. https://doi.org/10.1016/j.worlddev.2015.11.001.
[18] Bingnan Guo, Yu Wang, et.al. Impact of the digital economy on high-quality urban economic development: Evidence from Chinese cities[J]. Economic Modelling, Volume 120,2023(3).
- You need to elaborate on the overall methodology and data specifics.
Response: Thanks very much for the guidance of the Reviewer. Possibly we may not express clearly in the manuscript, and cause misunderstanding. The comments from reviewers are of great reference significance. We have supplemented the literature review of methods in several research methods section. The revisions are as following:
(1) We have supplemented some literature review in part 2.2.( Part 2.2, P6)
In China, scholars hold different views on the definition of the marine economy.[44] With the promotion of Industrial classification for ocean industries and their related activities (GB/T 20794-2006) in 2006, China’s marine economy has been defined officially as marine and marine-related industrial activities aimed at developing, utilizing and/or protecting the ocean. [45] All data were selected from China Ocean Yearbook and China Marine Statistical Yearbook from 2012 to 2019 to ensure temporal and spatial consistency in measurements. Digital economy, a brand-new economic form recently, originates from networked intelligence, which can be related back to the 1990s. [46] It is not clearly defined until the G20 leaders’ Hangzhou Summit in September 2016 that the digital economy, with information technology as the core and modern network as the carrier, comes from a series of economic activities intending at efficient application in communication technology and economic structure improvement. Currently, global economic governance is entering a new era through digital transformation. [47] Resultant business model innovations have fundamentally altered consumers’ expectations and behaviours, pressured traditional firms, and disrupted numerous markets. [48] As an emergent branch of economies, digital economy requires the building of a scientific knowledge base and represents the pattern transformation of economic growth in China and exerts a significant influence on industrial structure upgrading.[49]
[44] R. Zhao, S. Hynes, G.S. He, Defining and quantifying China’s ocean economy, Mar. Policy 43 (2014) 164–173.
[45] National Standardization Management Committee of China. Industrial classification for ocean industries and their related activities(GB/T 20794-2006). 2006.
[46] Chihiro Watanabe, Kashif Naveed, et.al. Measuring GDP in the digital economy: Increasing dependence on uncaptured GDP[J]. Technological Forecasting and Social Change, 2018(137):226-240. https://doi.org/10.1016/j.techfore.2018.07.053.
[47] Jinseop Jang, Jason McSarren, et.al. Global governance: present and future [J]. Palgrave Communications,2016(1):1-5. DOI: 10.1057/palcomms.2015.45.
[48] Nicolai J. Foss, Tina Saebi. Business models and business model innovation: Between wicked and paradigmatic problems[J]. Long Range Planning, 2018(2),51:9-21. https://doi.org/10.1016/j.lrp.2017.07.006.
[49] Wenrong Pan, Tao Xie,et.al. Digital economy: An innovation driver for total factor productivity [J]. Journal of Business Research, 139 (2022) 303–311. https://doi.org/10.1016/j.jbusres.2021.09.061.
- You need to align your results with prior literature.
Response: Thanks very much for the guidance of the Reviewer. The comments from reviewers are of great reference significance. Now we have carefully corrected and revised the conclusions for this revision. The revisions are as following:
First, the development level of the marine economy and digital economy in China’s coastal provinces and cities has shown an overall upward in time, but the overall level is low. In terms of spatial evolution, the development level of marine economy and digital economy both presents a spatial distribution pattern of “high in the East, and low in the South and North”. And the gap in the development level of digital economy between provinces is gradually widening.
Third, the coordination gap between regional marine economy and digital economy is obvious, showing a spatial distribution pattern of "high gap in the South and low gap in the East", but the gap within and between groups has gradually narrowed. The Theil index in the regional groups of the is divided into East Marine Economy Circle, North Marine Economy Circle and South Marine Economy Circle from small to large. The interregional index has dropped significantly from 15.1% to 11.8%, with an average annual decline of 2.1%. The coordinated development of regional marine economy and digital economy is transforming from low-quality, disordered and unbalanced to high-quality, sustainable and balanced.
- Inline 712- do your mean lack of data ...
Response: Thanks very much for the guidance of the Reviewer. The comments from reviewers are of great reference significance. Due to negligence, the expression of the manuscript is somewhat inaccurate, and we have modified it. The revisions are as following:
(1) Therefore, it is not only a scientific issue, but also a practical issue to explore how to promote the coordinated development of the marine economy and the digital economy from a multiple perspective and play a stronger synergy, which is of great significance to promote the sustainable development of the marine economy and the high-quality development of the digital economy. (P24)
(2) In addition, we proofread the full text and some spelling mistakes carefully. Thank you for your patient guidance and careful reading. Also, we have entrusted professional institutions to proofread and polish the language of the manuscript to improve the accuracy of the article.

Round 2
Reviewer 1 Report
The authors have made some changes, but have not fully responded to all the comments.
1. The authors never identified which industries make up the marine economy. Is it marine transportation, shipbuilding, ....? This needs to be reflected in the article. The same question remains open for the digital economy. The authors did not specify its structure, but the directions of the state policy in the field of digital economy.
2. When justifying the methods, the authors only described each method in more detail, but did not provide a comparison with other methods, did not indicate the criteria for selecting a method and did not conduct the necessary assessment of possible methods.
Author Response
Dear Reviewers:
Thank you for your letter concerning our manuscript entitled “Evolution of the coupling coordination between marine economy and digital economy” (sustainability: 2205818). The quality of this paper has been greatly improved after the pertinent opinions of experts and reviewers last time. Those comments are all valuable and very helpful for revising and improving our paper, as well as the important guiding significance to our researches. We have studied comments carefully and have made correction that we hope meet with approval. Revised portion are marked in red in the paper. (Please refer to the revised version for details.) The main corrections in the paper and the responds to the reviewer’s comments are as following. (Please refer to the page number of the revised version of the manuscript without comments.)
Responds to the reviewer 1’s comments:
The authors have made some changes, but have not fully responded to all the comments.
- The authors never identified which industries make up the marine economy. Is it marine transportation, shipbuilding, ....? This needs to be reflected in the article. The same question remains open for the digital economy. The authors did not specify its structure, but the directions of the state policy in the field of digital economy.
Response: Thanks very much for the guidance of the Reviewer. Possibly we may not express clearly in the manuscript, and cause misunderstanding. The comments from reviewers are of great reference significance. We have supplemented the composition of the marine economy and the digital economy in the article. The revisions are as following:
(1) We have already supplemented the composition of the marine economy and the digital economy in the second part of the article. (Part 2.1, P4, marked in yellow)
According to the National Standard of the People’s Republic of China “National Economic Industry Classification” (GB/T4754-2002) and the Marine Industry Standard of the People’s Republic of China “Marine Economic Statistical Classification and Code” (HY/T052-1999), the marine primary industry includes: marine fisheries; the marine secondary industry includes marine oil and gas industry, coastal sand mining, marine salt industry, marine chemical industry, the Marine biomedical industry, marine power and seawater utilization industry, marine ship industry, marine engineering construction industry, etc.; marine tertiary industry includes marine transportation industry, coastal tourism, marine scientific research, education, social services, etc. In a narrow sense, the marine economy includes five areas: production and services that obtain products directly from the sea, primary processing of the former, production and services that are directly applied to marine development, production and services based on seawater or the sea, and a range of marine-related scientific research, education and management. In a broad sense, marine industrial activities also include upstream and downstream industrial activities related to the marine economy in a narrow sense, regardless of whether the industry is located on the coast or not, as long as it provides certain conditions and a basis for the development and use of the sea.
While digital economy mainly includes communication industry, computer basic technology industry, software industry, software integration industry, Internet industry and other five industries. Digital economy includes digital industrialization and industry digitization. First, digital industrialization includes electronic information manufacturing industry, telecommunications industry, software and digital service industry, Internet industry, etc. Second, industrial digitalization means that traditional industries benefit from the improvement of production quantity and quality brought by digital technology. On this basis, the integration of various fields will be realized, resulting in new formats and new models. The development of digital economy should take new infrastructure as the core underlying architecture, including semiconductor equipment, communication facilities and services, software applications and other basic software and hardware. On this foundation, various Internet enterprises contact the specific application scenarios of enterprises and individuals, and build relevant digital application platforms to meet the specific needs of customers; In addition, the new output generated by the three traditional industries using digital technology will also constitute an important part of the digital economy, which is a direct manifestation of digital empowerment.
- When justifying the methods, the authors only described each method in more detail, but did not provide a comparison with other methods, did not indicate the criteria for selecting a method and did not conduct the necessary assessment of possible methods.
Response: Thanks very much for the guidance of the Reviewer. The comments from reviewers are of great reference significance. Due to the limitation of the layout, we have not discussed the choice of research methods in details in the article. The reasons and comparison are explained as follows:
(1) At present, the main models for measuring the coordination degree are the coupling coordination degree model and the Haken model. However, the Haken model is mainly used in the field of physics, and has certain limitations in the economic field. It is generally used to describe the external conditions. In the case of established conditions, the competition and synergy between the subsystems, parameters or factors within the system make the system develop qualitatively, and the model needs to be discretized. The coupling coordination degree model is a model of dynamic correlation between two or more systems affected by the interaction between themselves and the outside world to achieve coordinated development. It can reflect the degree of interdependence and mutual restraint between systems. It can also reflect the quality of coordination. It has been widely used in the economic and social fields. Therefore, the coupling coordination degree model is selected to measure the coordination degree of marine economy and digital economy.
(2) At present, coefficient of variation, Gini coefficient, Lorenz curve and Theil index are commonly used to analyze regional differences. The coefficient of variation mainly analyzes the degree of dispersion of the two groups of data, which is not restricted by data dimension and is relatively objective. However, this method mainly reflects the absolute value of the degree of data dispersion, which is not only affected by the degree of dispersion of variable values, but also by the average level of variable values. The Gini coefficient is calculated according to the area of Lorenz curve. Its biggest advantage is that it can intuitively grasp the size of inequality. However, Gini coefficient alone cannot essentially explain the status quo of intra-regional disparities. And one of the biggest advantages of the Theil index as a measure of inequality is that it measures the contribution of intra-group and inter-group disparities to the total gap. Regional differences in the coordination of the marine economy and the digital economy in coastal areas were measured to reflect not only inter-regional differences but also intra-regional development differences, and therefore the Thiel index was chosen to measure them.
(3) There are many methods for weight calculation, mainly as follows:
Factor analysis and principal component analysis can only obtain the weight of each factor, but cannot obtain the weight of each specific analysis item. It is necessary to combine other weight methods (usually entropy method) to obtain the weight of each specific item, and then get the final comprehensive result.
AHP hierarchy method and order graph method mainly use the relativity of value size for weight measurement, and information volume weighting method uses the coefficient of variation of data for weight assignment. These three methods are mainly used to evaluate objects when experts are scoring, which is highly subjective.
CRITIC weight method is a kind of objective weight method, which is mainly used to analyze when there is a certain correlation between indicators or factors, focusing on the analysis of data volatility and correlation. The independent weight method only considers the correlation between data, and the correlation number R value is obtained through regression analysis to indicate the collinearity strength.
Entropy method is an objective weight assignment method that uses entropy information to calculate the weight of each index combined with the variation degree of each index and provides a basis for comprehensive evaluation of multiple indexes. The main purpose of this paper is to measure the comprehensive level of marine economy and digital economy. Therefore, entropy method is selected for measurement, and the measured data results are objective and comprehensive.
In addition, we proofread the full text and some spelling mistakes carefully. Thank you for your patient guidance and careful reading. Also, we have entrusted professional institutions to proofread and polish the language of the manuscript to improve the accuracy of the article.

Reviewer 3 Report
Need not mention 'However, few scholars have examined such coordination' , in the intro. Please check for academic writing standards.
Author Response
Dear Reviewers:
Thank you for your letter concerning our manuscript entitled “Evolution of the coupling coordination between marine economy and digital economy” (sustainability: 2205818). The quality of this paper has been greatly improved after the pertinent opinions of experts and reviewers last time. Those comments are all valuable and very helpful for revising and improving our paper, as well as the important guiding significance to our researches. We have studied comments carefully and have made correction that we hope meet with approval. Revised portion are marked in red in the paper. (Please refer to the revised version for details.) The main corrections in the paper and the responds to the reviewer’s comments are as following. (Please refer to the page number of the revised version of the manuscript without comments.)
Responds to the reviewer 3’s comments:
- Need not mention 'However, few scholars have examined such coordination' , in the intro. Please check for academic writing standards.
Response: Thanks very much for the guidance of the Reviewer. The comments from reviewers are of great reference significance. To be more clearly and in accordance with the reviewer concerns, we made some revisions as follows: (Part 1, introduction , P3, marked in yellow)
In summary, most domestic and international scholars have used relevant measurement tools to analyze the role of digital technology and data elements in promoting the development of the marine economy and the coordinated development of the marine economy with the environment, science and technology, finance and the region. These research results have important reference value for this study. With the deep integration of marine economy and digital economy, it is necessary to conduct in-depth research on the coordination of marine economy and digital economy. Specifically, research has not been carried out on the state of the coordinated evolution between the marine economy and the digital economy, the spatial pattern of the evolution, the heterogeneity of the coordination between regions, and the factors affecting the coordination between the two. The comprehensive analysis of these problems is the urgent task for the high-quality development of marine economy and digital economy in coastal countries and regions in the new era.

Round 3
Reviewer 1 Report
The authors took into account all the recommendations.